# PerSEval: Assessing Personalization in Text Summarizers

**Sourish Dasgupta[1,*], Ankush Chander[1], Parth Borad[1], Isha Motiyani[1], Tanmoy Chakraborty[2,*]**
**[1]Dhirubhai Ambani Institute of Information & Communication Technology, India**
**[2]Indian Institute of Technology Delhi, India**
**Corresponding authors: `sourish_dasgupta@daiict.ac.in`, `tanchak@iitd.ac.in`**

## Abstract

Personalized summarization models cater to individuals' subjective understanding of saliency, as represented by their reading history and current topics of attention. Existing personalized text summarizers are primarily evaluated based on accuracy measures such as BLEU, ROUGE, and METEOR. However, a recent study argued that accuracy measures are inadequate for evaluating the *degree of personalization* of these models and proposed EGISES, the first metric to evaluate personalized text summaries. It was suggested that accuracy is a separate aspect and should be evaluated standalone. In this paper, we challenge the necessity of an accuracy leaderboard, suggesting that relying on accuracy-based aggregated results might lead to misleading conclusions. To support this, we delve deeper into EGISES, demonstrating both theoretically and empirically that it measures the degree of responsiveness, a necessary but not sufficient condition for degree-of-personalization. We subsequently propose `PerSEval`, a novel measure that satisfies the required sufficiency condition. Based on the benchmarking of ten SOTA summarization models on the PENS dataset, we empirically establish that – (i) `PerSEval` is reliable w.r.t human-judgment correlation (Pearson's $r = 0.73$; Spearman's $\rho = 0.62$; Kendall's $\tau = 0.42$), (ii) `PerSEval` has high rank-stability, (iii) `PerSEval` as a rank-measure is not entailed by EGISES-based ranking, and (iv) `PerSEval` can be a standalone rank-measure without the need of any aggregated ranking.

## 1  Introduction

With the incessant rise of information deluge, it has become even more imperative to develop efficient and accurate models to summarize the salient information in long documents for faster consumption, eliminating the irrelevant and supporting faster decision-making Ter Hoeve et al. (2022). However, the notion of saliency can be highly subjective in many use cases, particularly for documents containing multiple aspects and topics. This calls for summarizers that must be personalized to the users' preferences as depicted by their reading behavior and current topic(s) of attention (Ghodratnama et al., 2021; Ao et al., 2021). This calls for robust and reliable measures for evaluating the *degree-of-personalization* in them.

**Dearth of Personalization Evaluation.** Major studies on evaluating text summaries focus on accuracy measurement and include the proposal of a multitude of measures such as the ROUGE variants (e.g., ROUGE-$n$/L/SU4 etc.) (Lin, 2004), BLEU (Papineni et al., 2002), METEOR (Banerjee & Lavie, 2005), BERTScore (Zhang et al., 2020), PYRAMID (Gao et al., 2019) and the more recently proposed ones such as SUPERT (Gao et al., 2020), WIDAR (Jain et al., 2022), and InfoLM (Colombo et al., 2022b). Other studies acknowledged the need for more qualitative aspects such as consistency, coherence, and fluency (Yuan et al., 2021; Deng et al., 2021; Tam et al., 2023; Jain et al., 2023; Zhong et al., 2022). Recently, Vansh et al. (2023) established, both theoretically and empirically, that personalization is a different aspect than accuracy. The authors proposed a measure for degree-of-personalization for the first time and called it `EGISES` .

**The EGISES Paradox.** To put it succinctly, EGISES measures the average ratio of the (normalized) deviation between model-generated summaries and their corresponding user-expected summaries, capturing

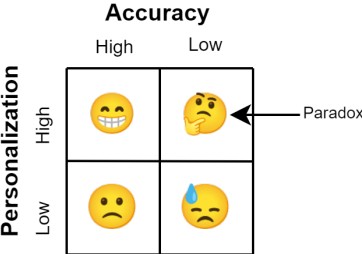

Figure 1: **EGISES Personalization-Accuracy Paradox.** The absurd case of high personalization (thereby high user-experience), yet low accuracy.

the strong notion of a model's degree of insensitivity to users' subjective expectations. However, we argue that this degree of insensitivity does not truly measure personalization but rather a related and necessary aspect, which can be understood as the *responsiveness* of a model. A high degree of personalization must imply a very high-quality user experience (UX). However, one cannot expect high UX while having low accuracy. At the same time, we demonstrate that a model can have high accuracy but a poor EGISES score. We resolve this apparent paradox by establishing both theoretically and empirically that EGISES, in reality, accounts for only the degree of responsiveness of models. In other words, we mathematically prove that a model can have a very good EGISES score but may still fall short of being personalized simply because of low accuracy performance.

**P-Accuracy is not a solution.** As an extension to standard accuracy measures, Vansh et al. (2023) proposed a measure called P-Accuracy (P-Acc) that penalizes accuracy by a factor that is a function of the EGISES score. Although P-Acc might apparently seem to address the EGISES paradox, in this paper, we prove that P-Acc fails to address the issue for two reasons. First, it has not been designed to evaluate the degree-of-personalization of a model but rather to get a more *realistic, human-evaluator adjustable accuracy score*. This makes P-Acc unable to distinguish the two models, where one exhibits low accuracy but high EGISES while the other exhibits high accuracy but low EGISES. Secondly, we prove that P-Acc is unstable for models with relatively low accuracy and may lead to anomalous results if we measure personalization using it. This motivates us to propose a novel consolidated personalization measurement framework for text summarizers, called **PerSEval** (**Per**sonalized **S**ummarizer **Eval**uator) that builds on the design principles of EGISES and bridges the "UX-gap".

**PerSEval Design Principle.** The underlying design principle of `PerSEval` entails that higher accuracy performance should not obfuscate the original EGISES score of a model. Otherwise, a model can be considered highly personalized simply because of its high accuracy, which is misleading, as was proven in (Vansh et al., 2023). However, the converse should not be true; hence, a lower accuracy score should penalize the original EGISES score. Based on these design objectives, we propose a penalty factor, called EDP (**E**ffective **D**EGRESS **P**enalty) that can be injected into EGISES to form `PerSEval` to measure the true degree of personalization. EDP incorporates the inconsistency of accuracy of model w.r.t its best accuracy performance and how much that is off from the maximum achievable accuracy (which is 0 for a normalized metric). In the best case, the EDP factor comes to 1 (i.e., the penalty is 0), while in the worst case, it tends to 0 (i.e., the penalty is 1).

**Observations and Insights.** We first empirically establish that the EGISES-based rank does not simply entail the `PerSEval`-based rank. In other words, models rank differently when EDP is applied. Therefore, we can conclude that the EGISES-paradox not just theoretically exists but has real evidence. We then show that `PerSEval` provides a much more reliable ranking of models with significantly higher human-judgment correlation in terms of Pearson's $r$ (0.73), Spearman's $\rho$ (0.62), and Kendall's $\tau$ (0.42). For fair comparisons, we consider the same top ten state-of-the-art news summarization models and the same PENS test dataset (Ao et al., 2021) that Vansh et al. (2023) considered to evaluate EGISES. We also take a step beyond (Vansh et al., 2023) and demonstrate that the accuracy leaderboard is not only insufficient but is at best redundant and at worst can be misleading for evaluating personalized summarizers. This is established by showing that

the Borda-Kendall consensus-based aggregated ranking (Cook & Seiford, 1982) of the models does not have a better human-judgement correlation when compared to the correlation of `PerSEval` alone.[1]

## 2 Background

### 2.1 `EGISES` is Not Enough: The Personalization-Accuracy Paradox

As proposed in (Vansh et al., 2023), the *degree-of-personalization* is a quantitative measure of how much a summarization model fine-tuned for personalization is adaptive to a user's (i.e., reader's) subjective expectation. This also implies that it measures how accurately a model can capture the *user's "evolving" profile reflected through **reading history*** (i.e., a temporal span of the reading and skipping actions of a user on a sequence of documents that is interleaved by the actions of generating and reading summaries). This is because the *subjective expectation is a function of the reading history*. **A low degree of personalization, by definition, implies poor user experience**. If a model does not efficiently capture the user's profile, it may lead to summaries that contain (a) additional irrelevant information, (b) are not coherent with the reading history, and (c) do not cover all the topics of interest within the reading history. In this situation, poor UX would mean that the user would have to spend more time getting to the information he/she is interested in or suffer from information overload and fatigue.

To illustrate this, we borrow the example given by Vansh et al. (2023) where we have two readers, Alice and Bob, having two very different reading histories over the same time span. Alice has been primarily reading news articles mostly on the broad topic of "*civilian distress*" in the Hamas-Israeli conflict, while Bob is following up on articles related to "*war-front battles*". In a situation where both Bob and Alice come up with an article covering both topics but rather wish to read the summary of the article instead, their respective expected summaries would be quite different. A non-personalized summarization model can be considered 100% accurate w.r.t recall and F-score-based measures (e.g., ROUGE-variants and METEOR, respectively) if it generates a summary that is a union of the expected summaries of Alice and Bob. On the other hand, a model can be considered 100% accurate w.r.t precision-based measures (eg., BLEU) if it covers the intersection between the expected summaries of Alice and Bob, which will not be much since they are quite different profiles. However, while the first situation would imply that Alice and Bob experience *significantly redundant content* in the generated summary that would drop their overall UX, the second case would entail that both Alice and Bob will have poor UX in terms of *significant lack of content coverage*. Hence, the utility of any summarization model as experienced by any user is directly associated with the subjective UX and cannot be traded in for accuracy, which is a rather clinical scoring mechanism that does not truly reflect the UX of a user in terms of redundancy and coverage (among other subjective aspects such as correctness and fluency).

This phenomenon was established theoretically and empirically by Vansh et al. (2023) for the first time, thereby emphasizing the fact that *accuracy measures can mislead an evaluator to select a model that has poor UX*. To address this, they proposed a novel measure, called EGISES, for personalization evaluation in summarizers. However, in this paper we establish both theoretically and empirically that if EGISES is used for personalization evaluation (i.e., a measure to understand a model's capacity to engage readers in terms of UX), then we can come to a rather paradoxical possibility where a model can have a high degree of personalization (i.e., acceptable EGISES score) but low accuracy, and yet, by definition, that would entail high UX (see Figure 1). In other words, although **high accuracy can lead to poor UX, the inverse (i.e., low accuracy leading to high UX) is absurd**. We term this as the **personalization-accuracy paradox** and attribute it to the incorrect interpretation or usage of EGISES. In this paper, we propose `PerSEval` as a **corrective** measure of EGISES that resolves this paradox[2]. In the following section, we show that EGISES measures *responsiveness*, a necessary yet distinct attribute to personalization.

---

[1]Code: https://github.com/KDM-LAB/Perseval-TMLR
[2]`PerSEval` should **not** be understood as an alternative "*improved*" measure, and therefore, is not comparable to EGISES.

## 2.2 Personalization vs. Responsiveness

In this section, we first distinguish *responsiveness* from *personalization*. Informally, responsiveness is the capacity of a model to discern the differences in the profiles (i.e., reading histories) of two readers quantitatively and accurately predict the dissimilarity in their corresponding expected summaries that is proportionate to their profile difference. However, there can be scenarios where **a model exhibits high responsiveness at the cost of losing accuracy**. To illustrate this, we continue with the example from the previous section. Suppose we observe an arbitrary model to generate two different summaries for a given news article, one focusing on "*Israeli Prime Minister*" and the other on "*Jewish protests on war*", skipping the article's content on civilian distress and war-front battle information. In that case, we have to conclude that the model apparently discerned the difference between Alice and Bob's profiles, thereby **predicting the proportionate dissimilarity** in the expected summaries but not the expected summaries themselves. Thus, the model is *inaccurate and yet responsive*. Therefore, interpreting such responsiveness as personalization leads us to the personalization-accuracy paradox. We prove this formally in Section 3.

We establish that EGISES measures how sensitive (or insensitive) a model is to the differences in the readers' subjective expectations (i.e., responsiveness) but not personalization. Therefore, EGISES can give a fairly good score to the model in the example. To elucidate this, we first define an Oracle personalized summarization model as follows:

**Definition 1. *Personalized Summarization Oracle.*** *A summarization model $M_{\boldsymbol{\theta},h}$ (parameterized with $\theta$) is an Oracle if for specific $j$-th reader profile $h_j$ (i.e., reading history) it generates an optimal summary $s^*_{(d_i,h_j)}$ of the document $d_i$ (i.e., $M_{\boldsymbol{\theta},h} : (d_i, h_j) \mapsto s^*_{(d_i,h_j)}$), where $s^*_{(d_i,h_j)} \equiv s^*_{u_{ij}} \equiv u_{ij}$; $u_{ij}$ is the $j$-th reader's **subjective** expected summary of $d_i$ and is determined by $h_j$.*

We now recall the notion of *insensitivity-to-subjectivity*, the foundation of EGISES, as in Vansh et al. (2023):

**Definition 2. *Weak Insensitivity-to-Subjectivity.*** *A summarization model $M_{\boldsymbol{\theta},h}$ is (weakly) Insensitive-to-Subjectivity w.r.t a given document $d_i$ and corresponding readers $j$ and $k$, if $\forall(u_{ij}, u_{ik})$, $(\sigma(u_{ij}, u_{ik}) \leq \tau^U_{max}) \iff (\sigma(s_{u_{ij}}, s_{u_{ik}}) > \tau^S_{max})$, where $\sigma$ is an arbitrary distance metric defined on the metric space $\mathcal{M}$, where $d, u$ and $s$ are defined[3], $\tau^U_{max}$ is the maximum limit for $u_i, u_j$ to be mutually indistinguishable, and $\tau^S_{max}$ is the maximum limit for $s_{u_i}, s_{u_j}$ to be mutually indistinguishable.*

**Definition 3. *Strong Insensitivity-to-Subjectivity.*** *A summarization model $M_{\boldsymbol{\theta},h}$ is (strongly) Insensitive-to-Subjectivity w.r.t a given document $d_i$ and corresponding readers $j$ and $k$, if $\forall(u_{ij}, u_{ik})$, $M_{\boldsymbol{\theta},h}$ satisfies: (i) the condition of weak insensitivity, and (ii) $(\sigma(u_{ij}, u_{ik}) > \tau^U_{max}) \iff (\sigma(s_{u_{ij}}, s_{u_{ik}}) \leq \tau^S_{max})$.*

Based on this notion, Vansh et al. (2023) defined (summary-level) "**deviation**" of a model $M_{\boldsymbol{\theta},h}$. We generalize this to our notion of summary-level ***Deg**ree-of-**R**esponsiven**ess*** (DEGRESS), the measure for responsiveness, as follows:

**Definition 4. *Summary-level* DEGRESS.** *Given a document $d_i$ and $j$-th reader's expected summary $u_{ij}$, the summary-level responsiveness of a personalized model $M_{\boldsymbol{\theta},h}$, (denoted by $\text{DEGRESS}(s_{u_{ij}}|(d_i, u_{ij}))$), is defined as the " **proportional divergence**" between model-generated summary $s_{u_{ij}}$ of $d_i$ for $j$-th user from all other user-specific summary versions w.r.t a corresponding divergence of $u_{ij}$ from all other user-profiles.*

$\text{DEGRESS}(s_{u_{ij}}|(d_i, u_{ij}))$ is formulated as follows:

$$\text{DEGRESS}(s_{u_{ij}}|(d_i, u_{ij})) = \frac{1}{|\mathbf{U}_{d_i}|} \sum_{k=1}^{|\mathbf{U}_{d_i}|} \frac{min(X_{ijk}, Y_{ijk}) + \epsilon}{max(X_{ijk}, Y_{ijk}) + \epsilon}$$

$$X_{ijk} = \frac{\exp(w(u_{ij}|u_{ik}))}{\sum_{l=1}^{|\mathbf{U}_{d_i}|} \exp(w(u_{ij}|u_{il}))} \cdot \sigma(u_{ij}, u_{ik}); \quad Y_{ijk} = \frac{\exp(w(s_{u_{ij}}|s_{u_{ik}}))}{\sum_{l=1}^{|\mathbf{U}_{d_i}|} \exp(w(s_{u_{ij}}|s_{u_{il}}))} \cdot \sigma(s_{u_{ij}}, s_{u_{ik}})$$

$$w(u_{ij}|u_{ik}) = \frac{\sigma(u_{ij}, u_{ik})}{\sigma(u_{ij}, d_i)}; \quad w(s_{u_{ij}}|s_{u_{ik}}) = \frac{\sigma(s_{u_{ij}}, s_{u_{ik}})}{\sigma(s_{u_{ij}}, d_i)}$$

$$\tag{1}$$

---

[3]$\sigma(u_i, u_i) = 0$; $\sigma(u_i, u_j) \in [0, 1]$; $\sigma$ satisfies positivity, reflexive, maximality, symmetry, and the triangle inequality.

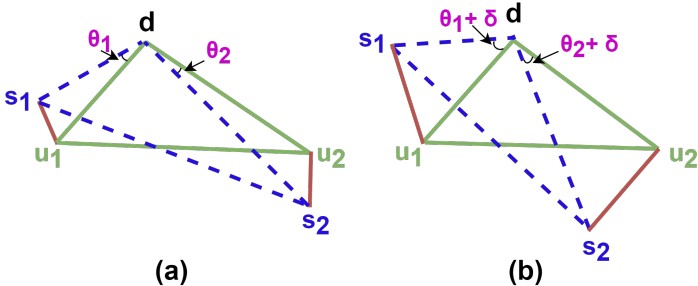

(a)         (b)

Figure 2: **Existence of EGISES Personalization-Accuracy Paradox:** High (and same; (a)) DEGRESS, yet low accuracy (red line; (b)).

Here, $|\mathbf{D}|$ is the total number of documents in the evaluation dataset, $|\mathbf{U}|$ is the total number of users who created gold-reference summaries that reflect their expected summaries (and thereby, their subjective preferences), and $|\mathbf{U}_{d_i}|$ $(= |\mathbf{S}_{d_i}|)$ is the number of users who created gold-references for document $d_i$. $w$ is the divergence of the model-generated summary $s_{u_{ij}}$ (and the corresponding expected summary $u_{ij}$) from document $d_i$ itself in comparison to all the other versions. It helps to determine how much percentage (therefore, the softmax function) of the divergence (i.e., $\sigma(s_{u_{ij}}, s_{u_{ik}})$ should be considered for the calculation of DEGRESS. If $s_{u_{ij}}$ is farther than $s_{u_{ik}}$ w.r.t $d_i$ then DEGRESS($s_{u_{ij}}|(d_i, u_{ij})$) < DEGRESS($s_{u_{ik}}|(d_i, u_{ik})$), implying that $M_{\boldsymbol{\theta},h}$ is more responsive to the $k$-th reader. A lower value of DEGRESS($s_{u_{ij}}|(d_i, u_{ij})$) indicates that while reader-profiles are different, the generated summary $s_{u_{ij}}$ is very similar to other reader-specific summaries (or vice versa), and hence, is not responsive at the summary-level. The system-level DEGRESS and EGISES have been formulated as follows:

$$\text{DEGRESS}(M_{\boldsymbol{\theta},h}) = \frac{\sum_{i=1}^{|\mathbf{D}|} \frac{\sum_{j=1}^{|\mathbf{U}_{d_i}|} \text{DEGRESS}(s_{u_{ij}}|(d_i, u_{ij}))}{|\mathbf{U}_{d_i}|}}{|\mathbf{D}|}; \quad \text{EGISES}(M_{\boldsymbol{\theta},h}) = 1 - \text{DEGRESS}(M_{\boldsymbol{\theta},h}) \tag{2}$$

EGISES measures the degree of **in**sensitivity-to-subjectivity for relative benchmarking of how much models *lack personalization* (i.e., a lower score is better within the range $[0,1]$) instead of assigning an absolute goodness score. As can be noted, the **EGISES formalism does not enforce any penalty on accuracy drop**. Here, accuracy would be an inverse function of $\sigma(s_{u_{ij}}, u_{ij})|d_i$ for the same metric distance $\sigma$ that DEGRESS uses. Hence, EGISES (and DEGRESS) should be interpreted as a measure of responsiveness (i.e., proportionate divergence) and not personalization.

## 3   EGISES Personalization-Accuracy Paradox: Formal Proof

In this section, we mathematically prove the existence of the condition that, for sufficiently high DEGRESS (and thereby EGISES), there exists low accuracy.

**Theorem 1.** *The accuracy $f^{-1}(\sigma(s_u, u))$ of a model $M_{\boldsymbol{\theta},h}$ on the metric space $\mathcal{M}$ with distance function $\sigma(\bullet, \bullet)$ can vary keeping* DEGRESS($s_u|(d, u)$) *constant.*

*Proof.* We follow the same triangulation proof technique as in (Vansh et al., 2023). Let $d, u$ and $s$ be triangulated as per Figure 2. Keeping $d$ and $u$ fixed, we can perform an arbitrary rotation operation with $d$ as center ($rot(\bullet, d, \delta)$; $\delta$: angle of rotation) on $s_{u_{ij}}$ and $s_{u_{ik}}$ s.t. $rot(\bullet, d, \delta)$ is a closure operator in $\mathcal{M}$. Now, $\exists (p, q) \in \mathcal{M}$, s.t.

$$\max_p \sigma(rot(s_{u_{ij}}, d_i, \delta_p), u_{ij}) > \min_p \sigma(rot(s_{u_{ij}}, u_{ij}, \delta_p), u_{ij});$$

$$\text{And similarly,} \max_q \sigma(rot(s_{u_{ik}}, d_i, \delta_q), u_{ik}) > \min_q \sigma(rot(s_{u_{ik}}, u_{ik}, \delta_q), u_{ik})$$

If $p = q$, then DEGRESS$(s_{u_{i\bullet}}|(d_i, u_{i\bullet}))$ (and therefore, EGISES) remains unchanged for a given $d_i$. However, due to the existence of a total ordering of accuracy $f^{-1}(\sigma(s, u))$, for any arbitrary $\alpha$, the accuracy, therefore, can be varied by changing $\delta$ from a minima to a maxima ($f^{-1}(\sigma(s, u)) \in [0, 1]$). $\qquad\square$

The proof establishes the theoretical existence of the personalization-accuracy paradox if we interpret EGISES as a measure of personalization instead of responsiveness. It sets the motivation to design a corrective measure for EGISES that truly measures personalization. As discussed in the introduction, Vansh et al. (2023) proposed a follow-up measure called P-Accuracy (P-Acc), that might seem to solve this problem. However, some major limitations of P-Acc prevent it from being used for evaluating degree-of-personalization. We elucidate that in the following section.

## 4 Limitations of P-Accuracy (P-Acc) as a Solution to the EGISES Paradox

In this section, we prove that P-Acc, as proposed in Vansh et al. (2023), cannot be used as a solution to the EGISES paradox. We begin with a recap of the formulation of P-Acc as follows:

$$P\text{-}Acc(M_{\boldsymbol{\theta},h}) = \text{Score}_{Acc}(M_{\boldsymbol{\theta},h}) \cdot \text{Unit}_{P\text{-}Acc}$$

$$\text{where: } \text{Unit}_{P\text{-}Acc} = 1 - [\alpha \cdot (\frac{f_{sig}(\beta \cdot \text{EGISES}(M_{\boldsymbol{\theta},h}))}{\text{Score}_{Acc}(M_{\boldsymbol{\theta},h})})]; \text{Score}_{Acc} : \text{Acc. score w.r.t a chosen accuracy measure}$$

$$(3)$$

Here, $\beta \in (0, 1]$ calibrates the importance to be given to lack of personalization in a model while $\alpha \in [0, 1]$ is the control that helps the human-evaluator to regulate the final penalty **to accuracy** due to lack of personalization. According to P-Acc, it *apparently* seems that there cannot be a situation where if both accuracy and EGISES scores are low (i.e., DEGRESS is high), the final score will be high, thereby preventing the EGISES Paradox. However, three limiting properties of P-Acc render it unusable for personalization evaluation. We prove their existence as follows:

**Theorem 2** (**Limitation 1**: Lack of Interpretability of Score). *There exists infinite possible model collections* $\Theta = \{\Theta_i \mid \Theta_i = \{M_{\boldsymbol{\theta}_{i\bullet},h}\}\}$ *where* $\forall(k,l)$ *pair* $: P\text{-}Acc(M_{\boldsymbol{\theta}_{ik},h}) = P\text{-}Acc(M_{\boldsymbol{\theta}_{il},h})$ *when (i)* $\text{Score}_{Acc}(M_{\boldsymbol{\theta}_{ik},h}) \gg \text{Score}_{Acc}(M_{\boldsymbol{\theta}_{il},h})$, *(ii)* $\text{Score}_{Acc}(M_{\boldsymbol{\theta}_{ik},h}) \ll \text{Score}_{Acc}(M_{\boldsymbol{\theta}_{il},h})$, *and (iii)* $\text{Score}_{Acc}(M_{\boldsymbol{\theta}_{ik},h}) = \text{Score}_{Acc}(M_{\boldsymbol{\theta}_{il},h})$.

*Proof.*

$$P\text{-}Acc(M_{\boldsymbol{\theta},h}) = \text{Score}_{Acc}(M_{\boldsymbol{\theta},h}) - \alpha \cdot f_{sig}(\beta \cdot \text{EGISES}(M_{\boldsymbol{\theta},h})), \quad \{\text{By definition}\}$$

$$\therefore \forall(k,l) \text{ pair} : P\text{-}Acc(M_{\boldsymbol{\theta}_{ik},h}) = P\text{-}Acc(M_{\boldsymbol{\theta}_{il},h})$$

$$\implies \text{Score}_{Acc}(M_{\boldsymbol{\theta}_{ik},h}) - \alpha \cdot f_{sig}(\beta \cdot \text{EGISES}(M_{\boldsymbol{\theta}_{ik},h})) = \text{Score}_{Acc}(M_{\boldsymbol{\theta}_{il},h}) - \alpha \cdot f_{sig}(\beta \cdot \text{EGISES}(M_{\boldsymbol{\theta}_{il},h}))$$

$$\implies \text{Score}_{Acc}(M_{\boldsymbol{\theta}_{ik},h}) - \text{Score}_{Acc}(M_{\boldsymbol{\theta}_{il},h}) = \alpha \cdot (f_{sig}(\beta \cdot \text{EGISES}(M_{\boldsymbol{\theta}_{ik},h})) - f_{sig}(\beta \cdot \text{EGISES}(M_{\boldsymbol{\theta}_{il},h})))$$

**Case 1/2:** Let us assume that a sufficiently large $k$ exists s.t. $\text{Score}_{Acc}(M_{\boldsymbol{\theta}_{ik},h}) = k \times \text{Score}_{Acc}(M_{\boldsymbol{\theta}_{il},h})$

$$\therefore f_{sig}(\beta \cdot \text{EGISES}(M_{\boldsymbol{\theta}_{ik},h})) - f_{sig}(\beta \cdot \text{EGISES}(M_{\boldsymbol{\theta}_{il},h})) = \frac{k-1}{\alpha} \cdot \text{Score}_{Acc}(M_{\boldsymbol{\theta}_{il},h})$$

$$\implies f_{sig}(\beta \cdot \text{EGISES}(M_{\boldsymbol{\theta}_{ik},h})) - f_{sig}(\beta \cdot \text{EGISES}(M_{\boldsymbol{\theta}_{il},h})) \geq k-1 \quad \{\because \alpha \in [0,1]\}$$

$$\implies \text{EGISES}(M_{\boldsymbol{\theta}_{ik},h}) - \text{EGISES}(M_{\boldsymbol{\theta}_{il},h}) \geq k-1$$

$$\implies \text{EGISES}(M_{\boldsymbol{\theta}_{ik},h}) = k' + \text{EGISES}(M_{\boldsymbol{\theta}_{il},h}) \quad \text{– which is a possible condition. Hence, the assumption is valid.}$$

**Case 3:** Let us assume $\text{Score}_{Acc}(M_{\boldsymbol{\theta}_{il},h}, h) = \text{Score}_{Acc}(M_{\boldsymbol{\theta}_{il},h}) \quad \{k = 1\}$

$$\therefore \text{EGISES}(M_{\boldsymbol{\theta}_{ik},h}, h) \geq \text{EGISES}(M_{\boldsymbol{\theta}_{il},h}, h) \quad \text{– which is a possible condition. Hence, the assumption is valid.}$$

$\qquad\square$

The proof entails that P-Acc scores cannot discriminate the three cases.

**Theorem 3** (**Limitation 2**: Lack of Interpretability of Lower Bound)**.** *There exists a condition when* $P\text{-}Acc(M_{\boldsymbol{\theta}_{i\bullet},h}) < 0$ *even when* $\text{Score}_{Acc}(M_{\boldsymbol{\theta}_{i\bullet},h}) \gg 0$.

*Proof.*

$$P\text{-}Acc(M_{\boldsymbol{\theta},h}) = \text{Score}_{Acc}(M_{\boldsymbol{\theta},h}) - \alpha \cdot f_{sig}(\beta \cdot \texttt{EGISES}(M_{\boldsymbol{\theta},h})), \quad \{\text{By definition}\}$$

$$\text{Let } \text{Score}_{Acc}(M_{\boldsymbol{\theta},h}) = k \ \text{ for sufficiently large } k$$

$$\text{Let us assume that } P\text{-}Acc(M_{\boldsymbol{\theta},h}) < 0$$

$$\therefore \text{Score}_{Acc}(M_{\boldsymbol{\theta},h}) - (\alpha \cdot f_{sig}(\beta \cdot \texttt{EGISES}(M_{\boldsymbol{\theta},h}))) < 0$$

$$\implies f_{sig}(\beta \cdot \texttt{EGISES}(M_{\boldsymbol{\theta},h})) = \delta_+ + \frac{\text{Score}_{Acc}(M_{\boldsymbol{\theta},h})}{\alpha} \quad \text{where } \delta_+ \text{ is a positive quantity}$$

$$\implies \texttt{EGISES}(M_{\boldsymbol{\theta},h}) = \frac{1}{\beta}\left(\ln\left(\frac{1 - (\delta_+ + \frac{\text{Score}_{Acc}(M_{\boldsymbol{\theta},h})}{\alpha})}{\delta_+ + \frac{\text{Score}_{Acc}(M_{\boldsymbol{\theta},h})}{\alpha}}\right)\right)$$

$$\therefore \exists \alpha, \beta, \delta_+, s.t. \ \text{Score}_{Acc}(M_{\boldsymbol{\theta},h}) \gg 0$$

$\square$

The proof implies that P-Acc can take a negative value even for a reasonably high accuracy model. As an example, for $\alpha = 0.5, \beta = 1$ (i.e., full importance to lack of personalization), $\texttt{EGISES}(M_{\boldsymbol{\theta},h}) \geq 0.9$ if $\text{Score}_{Acc}(M_{\boldsymbol{\theta},h}) \leq 0.35$ then P-Acc $< 0$.

**Theorem 4** (**Limitation 3**: Anomalous Behavior)**.** *There exists anomalous condition when* $P\text{-}Acc(M_{\boldsymbol{\theta}_{ik},h}) < P\text{-}Acc(M_{\boldsymbol{\theta}_{il},h})$ *even when* $\texttt{EGISES}(M_{\boldsymbol{\theta}_{ik},h}) < \texttt{EGISES}(M_{\boldsymbol{\theta}_{il},h})$ *(NT: lower is better).*

*Proof.*

$$\exists (k,l) \text{ pair}: P\text{-}Acc(M_{\boldsymbol{\theta}_{ik},h}) < P\text{-}Acc(M_{\boldsymbol{\theta}_{il},h}) < 0 \quad \{\text{From theorem 3}\}$$

$$\therefore \text{Score}_{Acc}(M_{\boldsymbol{\theta}_{ik},h}) - \text{Score}_{Acc}(M_{\boldsymbol{\theta}_{il},h}) < \alpha \cdot (f_{sig}(\beta \cdot \texttt{EGISES}(M_{\boldsymbol{\theta}_{ik},h}) - f_{sig}(\beta \cdot \texttt{EGISES}(M_{\boldsymbol{\theta}_{il},h}))$$

$$\implies f_{sig}(\beta \cdot \texttt{EGISES}(M_{\boldsymbol{\theta}_{ik},h}) - f_{sig}(\beta \cdot \texttt{EGISES}(M_{\boldsymbol{\theta}_{il},h}) > \frac{\text{Score}_{Acc}(M_{\boldsymbol{\theta}_{ik},h}) - \text{Score}_{Acc}(M_{\boldsymbol{\theta}_{il},h})}{\alpha}$$

$$\implies \texttt{EGISES}(M_{\boldsymbol{\theta}_{ik},h}) - \texttt{EGISES}(M_{\boldsymbol{\theta}_{il},h}) > \frac{\text{Score}_{Acc}(M_{\boldsymbol{\theta}_{ik},h}) - \text{Score}_{Acc}(M_{\boldsymbol{\theta}_{il},h})}{\alpha}$$

$$\Delta(\texttt{EGISES}(M_{\boldsymbol{\theta}_{ik},h}), \texttt{EGISES}(M_{\boldsymbol{\theta}_{il},h})) > \frac{1}{\alpha} \cdot \Delta(\text{Score}_{Acc}(M_{\boldsymbol{\theta}_{ik},h}), \text{Score}_{Acc}(M_{\boldsymbol{\theta}_{il},h})) - \text{which is a possible condition.}$$

$\square$

A bounding case of the proof condition is when for $\text{Score}_{Acc}(M_{\boldsymbol{\theta}_{ik},h}) = \text{Score}_{Acc}(M_{\boldsymbol{\theta}_{il},h}) = 0$, $\texttt{EGISES}(M_{\boldsymbol{\theta}_{ik},h}) = 0$ (**best case**) $\implies P\text{-}Acc(M_{\boldsymbol{\theta}_{ik},h}) = -1$ (*worst case*), and $\texttt{EGISES}(M_{\boldsymbol{\theta}_{il},h}) = 1$ (**worst case**) $\implies P\text{-}Acc(M_{\boldsymbol{\theta}_{il},h}) = -0.37$ (*better case*). Hence, P-Acc is not suitable for models that have relatively low accuracy scores. The fact that P-Acc is useful to regulate accuracy (and not measure personalization) is evident from Table 8 in Appendix C.2 where we observe that, unlike `PerSEval`-scores as outlined in section 8.1, P-Acc scores are heavily dominated by the evaluated PENS models' accuracy scores and therefore, do not differ significantly even though they lack responsiveness (i.e., low EGISES scores; see Table 7 in Appendix C.2). As a remedy, we propose `PerSEval` (**Per**sonalized **S**ummarizer **Eval**uator) in the next section.

## 5  PerSEval: Measure for Personalization

The design objective of `PerSEval` is two-fold: (i) to ensure that a model is penalized for poor accuracy performance, but at the same time, (ii) to ensure that the evaluation of responsiveness (i.e., DEGRESS) is not obfuscated by high accuracy (since high accuracy does not entail high responsiveness as proved in Vansh et al. (2023)). In other words, the underlying maxim behind the design of `PerSEval` is – *accuracy is not a reward, but lack of it is surely a penalty*! We term this penalty as *E**ffective DEGRESS Penalty Factor** (EDP). As per the design maxim, if accuracy is 100%, then there will be no EDP applied, and the `PerSEval` score will be the same as the DEGRESS score. In the subsequent sections, we develop the motivation and formulation of EDP.

### 5.1 Accuracy-drop Penalty (ADP)

In this section, we introduce the first component of EDP - ***A****ccuracy-****d****rop* ***P****enalty* (ADP). ADP is a document-level penalty due to a drop in accuracy for the best-case scenario where a model-generated summary of document $d_i$ ($s_{u_{ij}}$) is closest to the corresponding reader's expected summary $u_{ij}$. In this case, we denoted $s_{u_{ij}}$ as $s_{u_{i*}}$. We define ADP as follows:

**Definition 5.** ***Accuracy-drop Penalty.*** *Given document $d_i$ and user-generated summaries $\boldsymbol{U}_{d_i}$, the document-level* ADP *of a model $M_{\boldsymbol{\theta},h}$, denoted by* $\text{ADP}(s_{u_{i*}}|(d_i, u_{i*}))$, *is the relative deviation of the best performance $(\sigma^*(s_{u_{i\bullet}}, u_{i\bullet})|d_i)$ of $M_{\boldsymbol{\theta},h}$ $\forall \sigma(s_{u_{i\bullet}}, u_{i\bullet})|d_i$ from the best possible performance (i.e., $\boldsymbol{0}$) w.r.t its proximity to the worst possible performance (i.e., $\boldsymbol{1}$).*

Document-level ADP is formulated as follows:

$$\text{ADP}(s_{u_{i*}}|(d_i, u_{i*})) = \frac{1}{1 + 10^{\gamma \geq 4} \cdot \exp\left(-10 \cdot \frac{\sigma^*(s_{u_{i\bullet}}, u_{i\bullet})|d_i - \boldsymbol{0}}{(\boldsymbol{1} - \sigma^*(s_{u_{i\bullet}}, u_{i\bullet})|d_i) + \epsilon}\right)}$$

$$\text{where, } \sigma^*(s_{u_{i\bullet}}, u_{i\bullet})|d_i = \min_{j=1}^{|\mathbf{U}_{d_i}|} \sigma(s_{u_{ij}}, u_{ij})|d_i; \quad \{\epsilon : \text{An infinitesimally small number} \in (0, 1)\}$$

(4)

Here, ADP is defined as a shifted sigmoid function where $\gamma \geq 4$ helps to bring the minimum penalty to zero, while the factor of 10 in the exponentiation ensures that the maximum penalty reaches 1 when the ratio of the difference in the best case to the worst case is around 1.5 before it starts exploding (i.e., over-penalization). ADP ensures that even if the DEGRESS score is acceptable, a penalty due to accuracy drop can still be imposed as a part of EDP. ADP, however, fails to address the scenario where the best-case scenario is acceptable (i.e., accuracy is fairly high) but is rather an outlier case – i.e., for most of the other model-generated summary versions, there is a considerable accuracy drop. To address this issue, we introduce a second penalty component within EDP called ***A****ccuracy-in****c****onsistency* ***P****enalty* (ACP).

### 5.2 Accuracy-inconsistency Penalty (ACP)

ACP accounts for outlier conditions of the best performance, as explained previously. It evaluates how a model performs w.r.t accuracy for a specific generated summary compared to its average performance. More formally, it is defined as follows:

**Definition 6.** ***Accuracy-inconsistency Penalty.*** *Given document $d_i$ and user-generated summaries $\boldsymbol{U}_{d_i}$, the summary-level* ACP *of a model $M_{\boldsymbol{\theta},h}$, denoted by* $\text{ACP}(s_{u_{ij}}|(d_i, u_{ij}))$, *is defined as the relative deviation of the summary performance $\sigma(s_{u_{ij}}, u_{ij})|d_i$ of $M_{\boldsymbol{\theta},h}$ from its best performance $\sigma^*(s_{u_{i*}}, u_{i*})|d_i$ as compared to the deviation of its average performance $\overline{\sigma}(s_{u_{i\bullet}}, u_{i\bullet})|d_i$ from $\sigma^*(s_{u_{i*}}, u_{i*})|d_i$.*

It is to be noted that, unlike ADP, ACP is a summary-level measure. This penalty evaluates if the model consistently performs w.r.t accuracy and therefore, conversely, does not inject any additional penalty to DEGRESS when a model is consistent. The summary-level ACP is formulated as:

$$\text{ACP}(s_{u_{ij}}|(d_i, u_{ij})) = \frac{1}{1 + 10^{\gamma \geq 4} \cdot \exp\left(-10 \cdot \frac{\sigma(s_{u_{ij}}, u_{ij})|d_i - \sigma^*(s_{u_{i\bullet}}, u_{i\bullet})|d_i}{(\overline{\sigma}(s_{u_{i\bullet}}, u_{i\bullet})|d_i - \sigma^*(s_{u_{i\bullet}}, u_{i\bullet})|d_i) + \epsilon}\right)}$$

$$\text{where, } \overline{\sigma}(s_{u_{i\bullet}}, u_{i\bullet})|d_i = \frac{1}{|\mathbf{U}_{d_i}|} \sum_{j=1}^{|\mathbf{U}_{d_i}|} \sigma(s_{u_{ij}}, u_{ij})|d_i$$

(5)

### 5.3 PerSEval: Formulation

We now lay the design of the PerSEval framework as an extension to DEGRESS (i.e., $1 - $ EGISES). A multiplicative injection of EDP $\in (0, 1]$ should be such that the best accuracy (i.e., ADP $= 0$) with no inconsistency (i.e., ACP $= 0$) would lead to an EDP value of 1, and thereby, DEGRESS remains unobfuscated (which is the desired

objective). The following formulation of `PerSEval` guarantees these properties:

$$\text{PerSEval}(s_{u_{ij}}|(d_i, u_{ij})) = \text{DEGRESS}(s_{u_{ij}}|(d_i, u_{ij})) \times \text{EDP}(s_{u_{ij}}|(d_i, u_{ij}))$$

$$\text{where, } \text{EDP}(s_{u_{ij}}|(d_i, u_{ij})) = 1 - \frac{1}{1 + 10^{\alpha \geq 3} \cdot \exp\left(-(10^{\beta \geq 1} \cdot \text{DGP}(s_{u_{ij}}|(d_i, u_{ij})))\right)} \quad (6)$$

$$\text{and, } \text{DGP}(s_{u_{ij}}|(d_i, u_{ij})) = \text{ADP}(s_{u_{i*}}|(d_i, u_{i*})) + \text{ACP}(s_{u_{ij}}|(d_i, u_{ij}))$$

The system-level `PerSEval` score is as follows:

$$\text{PerSEval}(M_{\boldsymbol{\theta},h}) = \frac{\displaystyle\sum_{i=1}^{|\mathbf{D}|} \frac{\displaystyle\sum_{j=1}^{|\mathbf{U}_{d_i}|} \text{PerSEval}(s_{u_{ij}}|(d_i, u_{ij}))}{|\mathbf{U}_{d_i}|}}{|\mathbf{D}|} \quad (7)$$

We design `EDP` as an inverse sigmoid function of the overall ***DEGRESS Penalty*** (DGP), which sums up `ADP` and `ACP`. $\alpha$ and $\beta$ are hyper-parameters that help to control the shape of `EDP`. $\alpha \geq 3$ ensures that `EDP` is 1 (i.e., $1 - \mathbf{0}$) when there is no penalty, thereby making `PerSEval` equivalent to DEGRESS. $\beta \geq 1$ ensures that the function does not drop sharply, thereby over-penalizing (and hence, dampening) an otherwise fairly good DEGRESS score (i.e., responsiveness). Since $\beta$ may significantly affect the overall human-judgment correlation, we did an ablation study (Fig 3; Section 8.1) to find the optimal value (which was observed to be 1.7). The system-level `PerSEval` $\in [0, 1]$ and is bounded by the system-level `DEGRESS` score.

## 6 Benchmarking of SOTA Summarization Models w.r.t PerSEval

### 6.1 Model Benchmarking Dataset

**Microsoft PENS Dataset (News Domain).** Our study, as in (Vansh et al., 2023), assesses models using test data from the PENS dataset provided by Microsoft Research (Ao et al., 2021)[4]. This dataset pairs news headlines with articles, serving as concise summaries. The test set creation involved two phases: initially, 103 English speakers selected 50 articles of their interest from a pool of 1000, sorted based on exposure time. In the second phase, participants generated preferred headlines (gold references) for 200 articles without knowledge of the originals. The assignment ensured an average of four gold-reference summaries per article. The PENS dataset was chosen because it is the only one that contains the users' reading history, i.e., the temporal sequence of interactions (clicking, reading, and user-generated gold summaries), making it ideal for evaluating five SOTA personalized summarization models that require user reading history as an input.

**OpenAI CNN/Daily Mail Dataset (News Domain).** To understand the applicability of `PerSEval` on mainstream gold-standard news datasets, we design an **indirect evaluation methodology** with the OpenAI CNN/DM dataset (validation and test) released by Stiennon et al. (2020). The test dataset contains 639 articles processed by 19 policies, resulting in 5515 summaries rated by the same annotators on accuracy, coherence, coverage, and overall quality using a 7-point Likert scale. However, unlike the PENS dataset, this dataset lacks the **temporal span of human evaluator interactions forming the reading history** required by the PENS summarization models. Additionally, we do not have gold-reference summaries (but only ratings).[5] Hence, to conduct our experiments, we evaluated the personalization of 4 (out of 19) OpenAI policies, focusing on RLHF-tuned policies, as base-model and SFT-based policies are non-personalized. Since the **policy-generated summaries are not specific to any human annotator** as the policies are **not trained on any reading history**, we do not observe multiple user-specific summaries from the same policy - a necessary condition for evaluating personalization. To address this, we concatenate all summaries generated by different policies (say, $k$ such summaries) that were rated above 6 (out of $r_{max} = 7$) by a specific annotator (say, Alice) into one ***combined gold-reference summary*** as per Alice ($u_{\text{Alice}}$). After this

---

[4]We comply with the Microsoft Research License Terms.
[5]It is to be noted that human-written gold reference summaries were not always selected or rated as highest by the human annotators, **clearly indicating the subjectivity of saliency**.

step, the content of each of all the policy-generated summaries ($s$) that received some rating ($[1, r_{max}]$) from Alice ($r_{\text{Alice}}(s) \equiv r_{(s,\text{Alice})}$) is then concatenated ($k + (r_{max} - r_{(s,\text{Alice})})$)-times to create a policy-generated *surrogate personalized summary* ($s_{\text{Alice}}$) for Alice. Ideally, this surrogate should match Alice's combined gold-reference $u_{\text{Alice}}$. If $r_{(s,\text{Alice})}$ is much lower than $r_{max}$, then the surrogate should get penalized. The injection of redundant, unmatchable content helps to achieve this. We repeat this process for all annotators and generate the `PerSEval` scores of each policy across 27 (out of 639) articles. Since $k$ and $r_{(s,\bullet)}$ can differ between annotators, the penalty (and thereby length) of the surrogates will subjectively differ as well.

**OpenAI TL;DR (Reddit) Dataset (Open Domain).** To understand the broader applicability of `PerSEval`, we also appropriated the OpenAI TL;DR dataset Stiennon et al. (2020). This dataset is a collection of 123,169 Reddit posts adopted from the dataset by Völske et al. (2017) covering 29 subreddits (i.e., domains) of varied types, their corresponding OpenAI policy-generated summaries, and the human-written gold-reference summaries. A subset of the validation dataset comprises 1038 posts that were fed into 13 policies to generate 7713 summaries. Like the OpenAI CNN/DM dataset, 32 human annotators rated the summaries in the same format as in the previous OpenAI CNN/DM dataset. As a part of the appropriation, we carry the same method as that with CNN/DM leading to the evaluation of 4 RLHF-tuned policies (out of 13) and 57 (out of 1038) annotated articles. However, we would like to emphasize that this indirect benchmarking is **neither an equivalent evaluation methodology to the PENS dataset** based one, **nor is a simulation** of the required setup. There is so far no other dataset comparable to PENS for evaluating personalization, and hence, **all benchmarking results** in the outlined method should **be seen as baseline within the scope of the method**.

## 6.2 SOTA Summarization Models Evaluated

**Personalized Summarization Models (Microsoft PENS dataset).** We study ten SOTA summarization models for comparative benchmarking as in (Vansh et al., 2023). Five of them are specifically trained personalized models and follow the PENS framework (Ao et al., 2021): PENS-NRMS Injection-Type-1 (T1)/Type-2 (T2), PENS-NAML T1, PENS-EBNR T1 and PENS-EBNR T2. The others are generic SOTA models - BRIO (Liu et al., 2022), SimCLS (Liu & Liu, 2021), BigBird-Pegasus (Zaheer et al., 2020), ProphetNet (Qi et al., 2020), and T5-base (Orzhenovskii, 2021). The selections are based on their consistent top-5 ranking over the preceding four years on the CNN/Daily Mail news dataset Hermann et al. (2015). Appendix A contains model descriptions.

**Non-personalized Summarization Models (Microsoft PENS dataset).** For the non-personalized models, we follow the evaluation setup used by Vansh et al. (2023) by augmenting the documents with the reference summaries of each reader as document titles (i.e., cues). This results in subjective document versions corresponding to each reader. The models ideally should pick up the cues and generate them back as an output, thereby inducing an "apparent" sense of personalization. This injection process provides robust baselines for comparative evaluation.

**RLHF-tuned OpenAI Summarization Policies (OpenAI CNN/DM & TL;DR (Reddit) datasets).** We analyze five OpenAI policies (parameter size 1.3B and 6.7B) that have been RLHF-tuned using TL;DR (Reddit) training datasets. The base pre-trained model architecture is similar to that of GPT-3. The base model is then further fine-tuned in a supervised setup on a filtered set of the Reddit posts to generate SFT-models (denoted "*sup*"). The Reward Model (denoted "*rm*") is a linear head that predicts the rating. The Reinforcement Learning of the policy model is done via the PPO algorithm Zheng et al. (2023b).

## 6.3 Baseline Distance Metrics and Scores

`PerSEval` is a generic measurement framework (like EGISES) where the specific metric space $\mathcal{M}$ on which $\sigma$ is defined should be appropriately selected such that we achieve the best human-judgment (HJ) correlation. In this paper, we choose seven summarization accuracy metrics that are defined on standard algebraic spaces and plug them in `PerSEval` in isolation as distance metrics (i.e., $\sigma$) : (i) ROUGE (RG)-L, (ii) ROUGE (RG)-SU4,

| Basline-I: Microsoft PENS Test Dataset with Cue Injection | | | | | | | |
|---|---|---|---|---|---|---|---|
| **Models** | **PSE-RG-L** | **PSE-RG-SU4** | **PSE-METEOR** | **PSE-BLEU** | **PSE-JSD** | **PSE-BScore** | **PSE-InfoLM-$\alpha\beta$** |
| **BigBird-Pegasus** | **0.205** | **0.143** | **0.168** | **0.156** | **0.253** | **0.320** | **0.195** |
| **SimCLS** | 0.079 | 0.032 | 0.016 | 0.014 | 0.157 | 0.286 | 0.18 |
| **ProphetNet** | 0.054 | 0.027 | 0.018 | 0.016 | 0.097 | 0.228 | 0.135 |
| **T5 (Base)** | 0.035 | 0.022 | 0.011 | 0.016 | 0.073 | 0.207 | 0.124 |
| **BRIO** | 0.030 | 0.021 | 0.008 | 0.013 | 0.107 | 0.242 | 0.154 |
| Personalization Benchmarking: Microsoft PENS Test Dataset with Human Reading History | | | | | | | |
| **PENS-NAML T1** | 0.018 | 0.013 | 0.019 | 0.010 | 0.025 | 0.074 | 0.051 |
| **PENS-NRMS T1** | 0.016 | 0.011 | 0.016 | 0.008 | 0.022 | 0.062 | 0.045 |
| **PENS-EBNR T1** | 0.009 | 0.008 | 0.010 | 0.005 | 0.015 | 0.037 | 0.029 |
| **PENS-EBNR T2** | 0.002 | 0.005 | 0.003 | 0.002 | 0.006 | 0.008 | 0.013 |
| **PENS-NRMS T2** | 0.002 | 0.004 | 0.003 | 0.002 | 0.006 | 0.007 | 0.013 |
| Basline-II: OpenAI CNN/DM Test and TL;DR (Reddit) Validation Datasets with Human-Feedback (i.e., Rating) | | | | | | | |
| **Models (i.e., Policies)** | **PSE-RG-L** | **PSE-RG-SU4** | **PSE-METEOR** | **PSE-BLEU** | **PSE-JSD** | **PSE-BScore** | **PSE-InfoLM-$\alpha\beta$** |
| **sup4/rm4 (6.7B)** | **0.943/0.719** | **0.941/0.730** | **0.906**/0.525 | **0.887**/0.395 | **0.943**/0.732 | **0.998**/0.719 | **0.999/0.836** |
| **sup4/rm4-t.7 (6.7B)** | NA/0.711 | NA/0.648 | NA/**0.593** | NA/**0.465** | NA/**0.737** | NA/0.790 | NA/0.818 |
| **sup4/rm4 (1.3B)** | 0.617/0.523 | 0.617/0.502 | 0.413/0.357 | 0.612/0.163 | 0.617/0.566 | 0.994/**0.795** | **0.999**/0.605 |
| **sup4/rm4-kl14 (6.7B)** | 0.783/NA | 0.783/NA | 0.562/NA | 0.592/NA | 0.783/NA | 0.884/NA | 0.783/NA |
| **sup4/rm4-t.7 (1.3B)** | 0.505/0.481 | 0.5/0.414 | 0.267/0.417 | 0.266/0.323 | 0.502/0.523 | 0.92/0.776 | 0.646/0.602 |

Table 1: SOTA model-benchmarking on Microsoft PENS and OpenAI CNN/DM & TL;DR (Reddit) datasets w.r.t PerSEval (PSE) (**NA**: not applied on the dataset); `PerSEval` hyper-parameters: $\alpha = 3$, $\beta = 1.7$ (optimal $\beta$; see 3), and $\gamma = 4$. **Observation 1**: *Disagreement between keyword-based (RG-L, RG-SU4, METEOR, BLEU) and non-keyword-based PSE variants (JSD, BScore, InfoLM) in terms of non-personalized baseline leaderboard from rank-3 onward*; **Observation 2**: *InfoLM-$\alpha\beta$ has more reliable discriminatory performance with less sharp changes for PENS dataset*; **Observation 3**: *Specialized personalized models lack personalization w.r.t all variants of* `PerSEval`; **Observation 4**: *RLHF-tuned LLM-based policies exhibit much higher personalization baseline (i.e, baseline-II) compared to non-personalized specialized summarization models (i.e., baseline-I) within the limited scope of the surrogate-based indirect evaluation*; **Observation 5**: *PSE-scores are very high for top-performing OpenAI policy due to most surrogate summaries being highly rated and very similar to the combined gold-references.*

(iii) BLEU-1, (iv) METEOR,[6] (v) BertScore (BScore) defined on embedding space, (vi) Jenson-Shannon Distance (Menéndez et al., 1997) on probability space, and (vii) InfoLM-$\alpha\beta$ ($\alpha = 1$; $\beta = 1$) on probability space generated from the embedding space of a masked-LM. RG is chosen because it has a very high HJ (Pearson & Kendall) correlation ($> 0.7$) in most standard datasets such as CNN/DM and TAC-2008 (for RG-L) as reported in (Bhandari et al., 2020; Zhang et al., 2024), and DUC-2001/2002 (for RG-L/SU-4) (Lin, 2004). For the same reason, the $\alpha\beta$ variant of InfoLM is chosen. Comprehensive benchmark results w.r.t optimal `PerSEval` hyperparameters (i.e., $\alpha = 3$, $\beta = 1.7$, $\gamma = 4$; see Figure 3 for ablation results) for each of the variants are given in Table 1. We observe that most non-personalized models, such as BigBird-Pegasus, produce significantly stronger baselines across most `PerSEval` variants. However, keyword-based PSE variants (RG-L, RG-SU4, METEOR, BLEU) disagree with non-keyword-based PSE variants (JSD, BScore, InfoLM) in terms of the non-personalized baseline leaderboard from rank-3 onward. One reason for this is that JSD, BScore, and InfoLM being embedding-based can handle out-of-distribution (OOD) situations leading to lesser penalty for keyword-level mismatch that happens in abstraction summarization. We also find that the InfoLM-$\alpha\beta$ variant shows "*smoother discrimination*" (i.e., no sharp jump in the performance) when compared to RG-SU4, BLEU, and JSD. At the same time, the discriminatory performance of InfoLM-$\alpha\beta$ and JSD variants is much better than BScore, METEOR, and RG variants, leading to a more reliable leaderboard. The results on the OpenAI datasets should be seen within the context that: (a) the models are essentially policy **variants** that differs w.r.t size and specific PPO implementation, and (b) the indirect benchmarking method is limited (as outlined in section 6.2).

---

[6]First four defined as *similarity* functions on the string space while BScore is defined as *similarity* function (as opposed to distance function) on high dimensional embedding space; see Appendix B.1 for details on each of the seven measures.

# 7 Meta-evaluation of `PerSEval`: Experiment Design

In this section, we lay down the experiment design that forms the foundation to establish – (i) the **reliability** of `PerSEval` w.r.t human-judgment correlation, (ii) the **stability** of `PerSEval`, (iii) `PerSEval` as a rank-measure is **not entailed by `DEGRESS-rank`** (i.e., the EGISES paradox is empirically existent), and (iv) `PerSEval` can be a **standalone rank-measure** without the need of any aggregated ranking.

## 7.1 Meta-evaluation of `PerSEval` Reliability: Survey-based Direct Method

**Meta-evaluation Objective.** As pointed by Vansh et al. (2023), unlike the meta-evaluation of accuracy measures, direct meta-evaluation of personalization is not a feasible task since that would require human evaluators to work as a team and to understand how their subjective assessment of the `PerSEval` scores (i.e., to what extent they agree with the scores) corresponds with the differences in the model-generated summaries and their own subjective expected summaries. As an alternative, we propose a survey-based evaluation methodology to simulate this scenario. We argue that the meta-evaluation of `PerSEval` should have two objectives: (i) whether human evaluators would judge the responsiveness of models in the same "*ratio-way*" as what `DEGRESS` does, and (ii) whether human evaluators would apply the accuracy drop to the responsiveness in the same "*factor-way*" as what `PerSEval` does. In other words, the central objective is to validate to what extent human evaluators **agree with the design principles of `PerSEval` at a cognitive level**. **Participants.** We opened the survey to a selected pool of graduate students demonstrating fair English comprehension. The pool consisted of students from five different backgrounds: (i) computer science, (ii) electrical engineering, (iii) humanities & social sciences, (iv) mathematics, and (v) physics. The survey was promoted in ongoing courses, student associations, and social media groups. 169 students (∼45% male, ∼55% female) within the age group of 25-40 completed the survey. No other personal details were asked.

**Survey Procedure.** Each participant was shown a pair of gold-reference summaries (corresponding to two user profiles) for a specific news article from the PENS dataset. Along with this, five pairs of model-generated summaries were shown, each pair corresponding to five of the ten models studied in this paper (i.e., two participant responses covered all the ten models for a given document). To eliminate response bias, the participants were unaware of which of the six pairs was a gold-reference, and the model names were also not revealed. The same set (shuffled) was shown to two random participants to get an average. Each participant was asked to provide similarity ratings between 1 (low) and 6 (very high) for the summary pairs. A sample snapshot questionnaire is provided in Appendix D (figure 4).

**Modeling Human-judgment of Personalization (`PerSEval-HJ`).** We model a "*human version*" of `DEGRESS` (termed `DEGRESS`-HJ) using the normalized rating as the divergences (i.e., $\sigma(u_{ij}, u_{ik})$ and $\sigma(s_{u_{ij}}, s_{u_{ik}})$ for document $d_i$ and gold-references $u_{ij}$ and $u_{ik}$). As mentioned above, if `PerSEval` needs to be reliable, then the necessary condition is that `DEGRESS` should have a high correlation with `DEGRESS`-HJ, failing which it can be concluded that human evaluators do not interpret divergences in the ratio-way of `DEGRESS`. We use the standard Pearson's coefficient ($r$), Spearman's $\rho$, and Kendall's $\tau$ rank coefficients for this. As a sufficient part of the reliability test, we model the "*human version*" of EDP using standard accuracy measures (i.e., RG-L, RG-SU4, METEOR, BLEU, and InfoLM-$\alpha\beta$) as surrogates. The motivation behind this is that such measures are known to have high HJ-correlation. The objective was to check whether such a surrogate, if used as a factor (just as in `PerSEval`) with `PerSEval`-HJ, shows a high correlation with `PerSEval`, failing which implies that human evaluators do not cognitively resonate with this factor-styled discounting of `DEGRESS`

## 7.2 Meta-evaluation w.r.t Stability

`PerSEval`, being a rank measure, is said to be ***stable*** if, for any random sample of document (and corresponding gold-references) selected from the evaluation dataset, the rank of the evaluated models does not change. This meta-measure is important because it ***objectively*** checks if `PerSEval` can be relied on any arbitrary personalization evaluation dataset, unlike the `PerSEval`-HJ based reliability evaluation, which is subjective and indirect in nature. In order to establish stability, we need to define it w.r.t a specific sampling method.

**Sampling Method for Stability Meta-evaluation.** To understand the stability of `PerSEval`, we create random sample collections $\mathcal{C}^k_{(D,S,U)}$, where $k = \{80\%, 60\%, 40\%, 20\%\}$ of the PENS dataset and $N = |(D, S, U)|$.

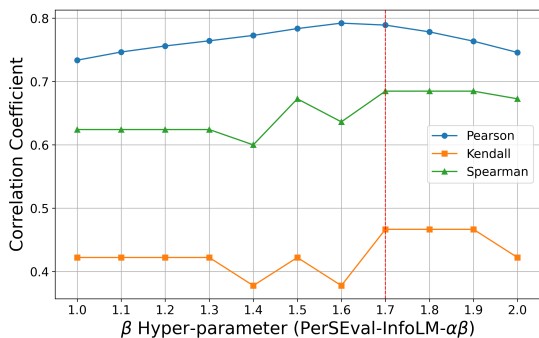

Figure 3: **PSE-ILM Ablation:** Effect of $\beta$ on HJ-Corr; Optimal performance at $\beta = 1.7$ across all three standard correlation measures (Pearson $r$, Spearman $\rho$, Kendall $\tau$).

Each collection has ten random sample sets $n_i^k \subset N$; $i = [1 : 10]$ (with replacement). We benchmark the models on all 40 sample sets to obtain corresponding leaderboards. We compare that with the rank obtained from the entire dataset. We formalize this notion of stability of a rank measure as follows:

**Definition 7.** ***Weakly Stable Rank Measure.*** *A rank measure is $\epsilon$-weakly stable if the maximum rank-correlation (w.r.t stat. $\tau$) between the measure-generated model-ranking on each $n_i^k$ and model-ranking on the entire dataset $N$ is $\leq \epsilon$.*

**Definition 8.** ***Strongly Stable Rank Measure.*** *A rank measure is $\delta$-strongly stable if (i) it is $\epsilon$-weakly stable, (ii) the bias over $\mathcal{C}_{(D,S,U)}$ w.r.t the mean score of each $\mathcal{C}_{(D,S,U)}^k$ for each evaluated model is $\leq \delta_b$, and (iii) the variance over $\mathcal{C}_{(D,S,U)}$ w.r.t the expected variance of the scores of each $\mathcal{C}_{(D,S,U)}^k$ for each model is $\leq \delta var$; $\delta = max(\delta_b, \delta_{var})$.*

## 8 Observations and Insights

In this section, we provide empirical support for the reliability and stability of `PerSEval` and show that accuracy leaderboards may be misleading (or, at best, redundant) for personalization analysis.

### 8.1 Meta-evaluation of `PerSEval`: Results

**Reliability of `PerSEval`.** We compute the HJ-correlation of the seven variants of `PerSEval` w.r.t to each of the five `PerSEval-HJ` variants (RG-L, RG-SU4, METEOR, BLEU, and InfoLM-$\alpha\beta$; see Table 2 for the results[7]). An 11-point hyper-parameter ablation study shows that the optimal correlation is at $\beta = 1.7$ (Figure 3). We observe that `PerSEval`$^{\beta=1.7}$-InfoLM-$\alpha\beta$ has the overall best correlation across all five `PerSEval-HJ` variants. We further observed that: (a) `PerSEval-HJ`[InfoLM$-\alpha\beta$] as a human-judgment estimate has the best performance of each `PerSEval`-variants (Pearson's $r = 0.79$; Spearman's $\rho = 0.68$; Kendall's $\tau = 0.47$ w.r.t PSE-HJ[InfoLM-$\alpha\beta$]), and (b) `PerSEval-InfoLM-`$\alpha\beta$ performs the best across. The high performance of `PerSEval`$^{\beta=1.7}$-InfoLM-$\alpha\beta$ can attributed to InfoLM-$\alpha\beta$, as an accuracy (and thereby, distance measure) having a very high system-level HJ-correlation (Pearson's $\tau \approx 0.95$; Kendall's $\tau \approx 0.85$) when compared to the other measures on the CNN/DM dataset, as reported in Colombo et al. (2022b).

**Indirect Evaluation Method is Inadequate for Meta-evaluation.** As discussed in section 6.2, the method of evaluating personalization of RLHF-tuned policies is indirect and limited in scope, since these policies are trained on annotator profile agnostic ratings and not on their reading history. Hence, although the OpenAI dataset may be used for generating baselines in this limited setup, they are not suitable for meta-evaluation of any personalization measure. To validate this point, we design an OpenAI dataset-oriented `PerSEval-HJ` variant where we model the divergence between policy-generated surrogate summaries

---

[7]`PerSEval-HJ`: Human judgment est.; Stat. Significance of Corr. (**Strong**, **Moderate**, **Low**, **None**): $p$-value $< 0.01$.

| Microsoft PENS dataset/OpenAI CNN-DM dataset/OpenAI Tl;DR (Reddit) dataset | | | | | | | |
|---|---|---|---|---|---|---|---|
| **PerSEval-HJ$^{\text{RG-L}}$** | | | | | | | |
| **HJ-Corr.** | **PSE-RG-L** | **PSE-RG-SU4** | **PSE-METEOR** | **PSE-BLEU** | **PSE-JSD** | **PSE-BScore** | **PSE-InfoLM-$\alpha\beta$** |
| Pearson's $r$ | **0.23**/0.90/-0.89 | **0.34**/0.89/-0.81 | **0.49**/0.89/-0.83 | **0.43**/0.67/-0.63 | **-0.05**/0.90/-0.92 | **-0.41**/0.02/0.07 | **0.79**/0.21/-0.71 |
| Spearman's $\rho$ | **-0.25**/0.80/-0.80 | **-0.27**/0.80/-0.80 | **0.15**/0.80/-0.80 | **-0.28**/0.40/-0.80 | **-0.26**/0.80/-1.00 | **-0.28**/0.20/0 | **0.69**/0.40/-0.80 |
| Kendall's $\tau$ | **-0.20**/0.67/-0.67 | **-0.24**/0.67/-0.67 | **0.16**/0.67/-0.67 | **-0.24**/0.33/-0.67 | **-0.20**/0.67/-1.00 | **-0.24**/0/0 | **0.47**/0.33/-0.67 |
| **PerSEval-HJ$^{\text{RG-SU4}}$** | | | | | | | |
| **HJ-Corr.** | **PSE-RG-L** | **PSE-RG-SU4** | **PSE-METEOR** | **PSE-BLEU** | **PSE-JSD** | **PSE-BScore** | **PSE-InfoLM-$\alpha\beta$** |
| Pearson's $r$ | **0.91**/0.90/-0.89 | **0.96**/0.89/-0.81 | **0.98**/0.67/-0.82 | **0.98**/0.89/-0.62 | **0.77**/0.89/-0.92 | **0.49**/0.03/0.07 | **0.73**/0.21/-0.70 |
| Spearman's $\rho$ | **0.07**/0.80/-0.80 | **0.05**/0.80/-0.80 | **-0.10**/0.80/-0.80 | **-0.02**/0.40/-0.80 | **0.16**/0.80/-1.00 | **0.15**/0.20/0 | **0.77**/0.40/-0.80 |
| Kendall's $\tau$ | **0.11**/0.67/-0.67 | **0.07**/0.67/-0.67 | **-0.07**/0.33/-0.67 | **-0.02**/0.67/-0.67 | **0.20**/0.67/-1.00 | **0.16**/0/0 | **0.6**/0.33/-0.67 |
| **PerSEval-HJ$^{\text{METEOR}}$** | | | | | | | |
| **HJ-Corr.** | **PSE-RG-L** | **PSE-RG-SU4** | **PSE-METEOR** | **PSE-BLEU** | **PSE-JSD** | **PSE-BScore** | **PSE-InfoLM-$\alpha\beta$** |
| Pearson's $r$ | **0.06**/0.90/-0.90 | **0.18**/0.90/-0.81 | **0.36**/0.89/-0.86 | **0.29**/0.67/-0.67 | **-0.23**/0.90/-0.92 | **-0.56**/0.02/0.08 | **0.77**/0.21/-0.72 |
| Spearman's $\rho$ | **-0.28**/0.80/-0.80 | **-0.33**/0.80/-0.80 | **0.12**/0.80/-0.80 | **-0.27**/0.40/-0.80 | **-0.32**/0.80/-1.00 | **-0.37**/0.20/0 | **0.71**/0.40/-0.80 |
| Kendall's $\tau$ | **-0.29**/0.67/-0.67 | **-0.33**/0.67/-0.67 | **0.07**/0.67/-0.67 | **-0.24**/0.33/-0.67 | **-0.38**/0.67/-1.00 | **-0.42**/0/0 | **0.51**/0.33/-0.67 |
| **PerSEval-HJ$^{\text{BLEU}}$** | | | | | | | |
| **HJ-Corr.** | **PSE-RG-L** | **PSE-RG-SU4** | **PSE-METEOR** | **PSE-BLEU** | **PSE-JSD** | **PSE-BScore** | **PSE-InfoLM-$\alpha\beta$** |
| Pearson's $r$ | **0.84**/0.90/-0.89 | **0.91**/0.89/-0.81 | **0.95**/0.89/-0.82 | **0.95**/0.67/-0.62 | **0.67**/0.89/-0.92 | **0.36**/0.03/0.07 | **0.63**/0.21/-0.71 |
| Spearman's $\rho$ | **-0.13**/0.80/-0.80 | **-0.12**/0.80/-0.80 | **-0.08**/0.80/-0.80 | **-0.19**/0.40/-0.80 | **-0.03**/0.80/-1.00 | **-0.02**/0.20/0 | **0.65**/0.40/-0.80 |
| Kendall's $\tau$ | **-0.16**/0.67/-0.67 | **-0.11**/0.67/-0.67 | **-0.07**/0.67/-0.67 | **-0.20**/0.33/-0.67 | **-0.07**/0.67/-1.00 | **-0.02**/0/0 | **0.51**/0.33/-0.67 |
| **PerSEval-HJ$^{\text{JSD}}$** | | | | | | | |
| **HJ-Corr.** | **PSE-RG-L** | **PSE-RG-SU4** | **PSE-METEOR** | **PSE-BLEU** | **PSE-JSD** | **PSE-BScore** | **PSE-InfoLM-$\alpha\beta$** |
| Pearson's $r$ | **0.54**/0.90/-0.89 | **0.59**/0.90/-0.80 | **0.64**/0.89/-0.86 | **0.62**/0.67/-0.68 | **0.43**/0.90/-0.92 | **0.12**/0.01/0.06 | **0.43**/0.21/-0.72 |
| Spearman's $\rho$ | **0.02**/0.80/-0.80 | **-0.02**/0.80/-0.80 | **-0.07**/0.80/-0.80 | **-0.11**/0.40/-0.80 | **0.11**/0.80/-1.00 | **0.07**/0.20/0 | **0.50**/0.40/-0.80 |
| Kendall's $\tau$ | **0.01**/0.67/-0.67 | **-0.04**/0.67/-0.67 | **-0.08**/0.67/-0.67 | **-0.15**/0.33/-0.67 | **0.14**/0.67/-1.00 | **0.09**/0/0 | **0.38**/0.33/-0.67 |
| **PerSEval-HJ$^{\text{BScore}}$** | | | | | | | |
| **HJ-Corr.** | **PSE-RG-L** | **PSE-RG-SU4** | **PSE-METEOR** | **PSE-BLEU** | **PSE-JSD** | **PSE-BScore** | **PSE-InfoLM-$\alpha\beta$** |
| Pearson's $r$ | **0.65**/0.93/-0.76 | **0.59**/0.92/-0.66 | **0.47**/0.89/-0.73 | **0.51**/0.70/-0.52 | **0.81**/0.93/-0.80 | **0.83**/-0.04/-0.17 | **0.56**/0.24/-0.50 |
| Spearman's $\rho$ | **0.60**/0.80/-0.80 | **0.56**/0.80/-0.80 | **0.24**/0.80/-0.80 | **0.47**/0.40/-0.80 | **0.69**/0.80/-1.00 | **0.64**/0.20/0 | **0.84**/0.40/-0.40 |
| Kendall's $\tau$ | **0.72**/0.67/-0.67 | **0.71**/0.67/-0.67 | **0.25**/0.67/-0.67 | **0.66**/0.33/-0.67 | **0.82**/0.67/-1.00 | **0.81**/0/0 | **0.73**/0.33/-0.33 |
| **PerSEval-HJ$^{\text{InfoLM}-\alpha\beta}$** | | | | | | | |
| **HJ-Corr.** | **PSE-RG-L** | **PSE-RG-SU4** | **PSE-METEOR** | **PSE-BLEU** | **PSE-JSD** | **PSE-BScore** | **PSE-InfoLM-$\alpha\beta$** |
| Pearson's $r$ | **0.49**/0.91/-0.76 | **0.60**/0.90/-0.71 | **0.66**/0.88/-0.61 | **0.58**/0.67/-0.35 | **0.78**/0.90/-0.79 | **0.51**/-0.04/-0.10 | **0.76**/0.20/-0.67 |
| Spearman's $\rho$ | **0.21**/0.80/-0.40 | **0.60**/0.80/-0.40 | **0.50**/0.80/-0.40 | **0.55**/0.40/-0.40 | **0.62**/0.80/-0.80 | **0.47**/0.20/-0.60 | **0.61**/0.40/-0.40 |
| Kendall's $\tau$ | **0.24**/0.67/-0.33 | **0.51**/0.67/-0.33 | **0.42**/0.67/-0.33 | **0.47**/0.33/-0.33 | **0.45**/0.67/-0.67 | **0.38**/0/-0.33 | **0.42**/0.33/-0.33 |

Table 2: **PerSEval (PSE) Reliability:** Human-judgment (HJ) corr. between PSE$^{\beta=1.7}$-X and PerSEval-HJ-X on Microsoft PENS/OpenAI CNN-DM/OpenAI TL;DR (Reddit) datasets. **Observation 1**: PerSEval-*InfoLM-$\alpha\beta$ has maximum highest HJ correlation across all* PerSEval-*HJ variants on PENS dataset;* **Observation 2**: PerSEval-*HJ-InfoLM-$\alpha\beta$ and* PerSEval-*HJ-BScore are optimal meta-evaluation measures since they are most closely aligned to all the seven* PerSEval *variants;* **Observation 3**: *Difference in human annotator ratings does not serve as reliable surrogate for estimating* DEGRESS-*HJ (and thereby,* PerSEval-*HJ) as evident from* contradictory results *on structurally equivalent OpenAI CNN/DM and TL;DR (Reddit) datasets;* **Observation 4**: *As a follow-up, the indirect evaluation method can serve for baseline generation but not meta-evaluation.*

($\sigma(s_{u_{ij}}, s_{u_{ik}})$) with the difference in the ratings given by the annotators (i.e., $|r(s_{u_{ij}}) - r(s_{u_{ik}})|$) and that between combined gold-references ($\sigma(u_{ij}, u_{ik})$) with the difference in the average ratings of all summaries included in the combined gold reference (i.e., $|\bar{r}(u_{ij}) - \bar{r}(u_{ik})|$). For calculating penalty due to EDP, we take the distance between gold-reference and generated summary to be $|r(s_{u_{ij}}) - \bar{r}(u_{ij})|$). We then create seven variants of PerSEval-HJ-X as in the PENS dataset-based evaluation, where X stands for the measure used for estimating the DEGRESS normalization factor (i.e., the distance of $s_{u_{i\bullet}}$ and $u_{i\bullet}$ from the document $d_i$; see equation 2). It is evident from Table 2 that the HJ-correlation so computed is unstable since the results are contradictory even though both the OpenAI datasets are structured in exactly the same manner with four common policies. This shows that the difference in rating cannot be an alternative to the direct similarity judgment that was collected via the survey.

| Inter-Corr. | EG-RG-L | EG-RG-SU4 | EG-METEOR | EG-BLEU | EG-JSD | EG-BScore | EG-InfoLM-$\alpha\beta$ |
|---|---|---|---|---|---|---|---|
| Spearman's $\rho$ | 0.903 | 0.9758 | 0.8667 | 0.9636 | 0.997 | 0.8476 | 0.9879 |
| Kendall's $\tau$ | 0.8222 | 0.9111 | 0.7333 | 0.9111 | 0.9888 | 0.7047 | 0.9555 |

Table 3: **EGISES (EG-X) Paradox:** PSE-X$^{\beta=1.7}$ rank-disagreement ($< 1$ inter-correlation) due to `EDP`. See Table 7 in Appendix C.2 for detailed EGISES scores.

| | | PENS Test Dataset Sample Set (Random Selection) | | | | | | |
|---|---|---|---|---|---|---|---|---|
| **Ranking** | **Models** | **100%** | **80%** | **60%** | **40%** | **20%** | **Bias** | **Variance** |
| 1 | BigBird-Pegasus | **0.15** | **0.153** | **0.1535** | **0.1558** | **0.1564** | 0.0024 | 5.81E-06 |
| 2 | SimCLS | 0.133 | 0.1351 | 0.1356 | 0.1357 | 0.138 | 0.0018 | 3.08E-06 |
| 3 | BRIO | 0.112 | 0.1143 | 0.1151 | 0.116 | 0.1155 | 0.0013 | 1.77E-06 |
| 4 | ProphetNet | 0.098 | 0.1003 | 0.1018 | 0.1012 | 0.1031 | 0.0017 | 2.90E-06 |
| 5 | T5 (Base) | 0.088 | 0.0899 | 0.0905 | 0.091 | 0.0912 | 0.0012 | 1.33E-06 |
| 6 | PENS-NAML T1 | 0.036 | 0.0364 | 0.0367 | 0.0376 | 0.039 | 0.0012 | 1.41E-06 |
| 7 | PENS-NRMS T1 | 0.0315 | 0.0326 | 0.0327 | 0.0331 | 0.033 | 0.0006 | 3.26E-07 |
| 8 | PENS-EBNR T1 | 0.0206 | 0.0209 | 0.021 | 0.0212 | 0.0228 | 0.0008 | 6.00E-07 |
| 9 | PENS-NRMS T2 | 0.0103 | 0.0103 | 0.0102 | 0.0107 | 0.0111 | 0.0003 | 1.14E-07 |
| 10 | PENS-EBNR T2 | 0.0096 | 0.0097 | 0.0097 | 0.0103 | 0.0107 | 0.0004 | 1.84E-07 |

Table 4: **PerSEval$^{\beta=1.7}$-InfoLM-$\alpha\beta$ Stability**: 0.0024-strongly-stable w.r.t $\epsilon$-Spearman $= 1$; $\epsilon$-Kendall $= 1$.

| HJ-Corr. | PSE-ILM-$\alpha\beta$ | BK(PSE-ILM-$\alpha\beta$, RG-L) | BK(PSE-LM-$\alpha\beta$, ILM-$\alpha\beta$) | BK(PSE-ILM-$\alpha\beta$, BLEU) |
|---|---|---|---|---|
| Spearman $\rho$ | **0.6849** | 0.1656 | -0.3447 | 0.632 |
| Kendall $\tau$ | **0.4667** | 0.0698 | -0.3027 | 0.46 |

Table 5: **Accuracy-leaderboards may mislead**: HJ-Corr. of Borda-Kendall (BK) consensus-based aggregated rank vs. PSE$^{\beta=1.7}$-InfoLM (ILM)-$\alpha\beta$.

**Evidence of the EGISES Paradox.** We look at the inter-correlations between EGISES-rank and `PerSEval`-rank of the selected models on the PENS dataset. The values less than 1 across all the variants (see Table 3) suggest that `PerSEval` is not just an offset of EGISES and is not entailed by it (which would have otherwise been the case if the EGISES paradox was non-existent in reality).

**Stability of `PerSEval`.** We compute $\epsilon$-$\delta$-stability of the best performing `PerSEval`-InfoLM-$\alpha\beta$ on PENS over the ten models as per the sampling method and stability definitions in Section 7.2. We observe `PerSEval` to be 0.0024[8]-strongly-stable w.r.t Spearman-$\epsilon = 1$ and Kendall-$\epsilon = 1$ (see Table 4). This establishes a very high stability of `PerSEval` along with its reliability, making it robust. For detailed results, see Appendix C.1.

## 8.2 Accuracy Leaderboards may Mislead

We demonstrate that accuracy leaderboards, at best, are redundant and `PerSEval`-rank is sufficient to capture personalization. For this, we generate the Borda-Kendall consensus-based aggregated rank (Colombo et al., 2022a) and compare the HJ-correlation with that of `PerSEval`-InfoLM-$\alpha\beta$. We observe that the stand-alone HJ correlation has the same strength w.r.t the aggregated rank for accuracy measures like BLEU, thereby rendering them redundant in the context of personalization evaluation, while is significantly higher (Spearman $\rho$: 0.51+↑; Kendall $\tau$: 0.40+↑) than that of measures such as RG-L, InfoLM-$\alpha\beta$ (see Table 5). This indicates that accuracy ranking can also inject noise.

## 8.3 Computational Complexity of `PerSEval`.

We now show that the best performing `PerSEval`-$\alpha\beta$ is linear time w.r.t to summary lengths - i.e., $O(l_s + l_u)$ for a fixed evaluation dataset $|\mathbf{D}|$ where $l_{s_u}$ and $l_u$ are lengths of model-generated summary $s_u$ and user-generated reference summary $u$.

---

[8]Maximum of $\delta$-bias and $\delta$-variance over all ten models

**Theorem 5** (`PerSEval` Time Complexity)**.** *The best worst-case time complexity of* `PerSEval` *is* $O\left(|\boldsymbol{D}| \cdot (l_{s_u} + l_u)\right).$

*Proof.* Let $t_{\sigma_X}$ be the time-complexity of computing the distance using measure $\sigma_X$.[9]

$O(\texttt{DEGRESS}(s_{u_{ij}}|(d_i, u_{ij}))) = O(|\mathbf{U}_{d_i}| \cdot t_{\sigma_X}) = O(t_{\sigma_X})$

$\{\because |\mathbf{U}_{d_i}| = \text{number of reference summaries per article} \leq \text{constant } C\};$

$\implies O(\texttt{DEGRESS}) = |\mathbf{D}| \cdot (|\mathbf{U}_{d_i}| \cdot O(\texttt{DEGRESS}(s_{u_{ij}}|(d_i, u_{ij})))) = |\mathbf{D}| \cdot O(t_{\sigma_X})$

Now, $O(\texttt{EDP}(s_{u_{ij}}|(d_i, u_{ij})) = (O(\texttt{ADP}(s_{u_{ij}}|(d_i, u_{ij}))) + O(\texttt{ACP}(s_{u_{ij}}|(d_i, u_{ij})))) = O(\mathbf{U}_{d_i} \cdot t_{\sigma_X}) = O(t_{\sigma_X})$

$\implies O(\texttt{PerSEval}) = |\mathbf{D}| \cdot O(t_{\sigma_X}) = O(|\mathbf{D}| \cdot t_{\sigma_X})$

For $\sigma_X \in \{\text{RG-L, RG-SU4, JSD, InfoLM-}\alpha\beta\}: O(t_{\sigma_X}) = O(l_{s_u} + l_u);$

{Suffix-tree based matching for RG;}

{No. of layers, dimensions, and vocabulary size for a fixed MLM (BERT) used in InfoLM can be taken as constant}

For $\sigma_X \in \{\text{BLEU, METEOR, BScore}\}: O(t_{\sigma_X}) = O(l_{s_u} \cdot l_u);$

{No. of layers, dimensions, and vocabulary size for a fixed BERT model used in BScore can be taken as constant}

$\therefore O(\texttt{PerSEval}) = O\left(|\mathbf{D}| \cdot (l_{s_u} + l_u)\right)$ for the best-performing `PerSEval-`$\alpha\beta$

$\square$

For computing `PerSEval`, we observe an average runtime of 2:37 minutes for each of the evaluated ten models on the PENS dataset (3840 articles), which is reasonable for an offline procedure.[10]

# 9  Related Work

**Evaluation of Personalization**  Personalization evaluation has been well studied in recommendation systems (recsys) (Zangerle & Bauer, 2022), such as metrics based on the Jaccard Index, rank-order edit distance (Hannak et al., 2013), MAE/RMSE/Hit-Ratio (Li et al., 2024), and nDCG (normalized Discounted Cumulative Gain) (Matthijs & Radlinski, 2011). A comprehensive compilation of all recsys-oriented metrics and their applications can be found in (Zangerle & Bauer, 2022; Kuanr & Mohapatra, 2021). While relevant to recsys, these metrics are not pertinent for text summarization since they rely on human feedback (such as clicks and likes) on a rank list of potentially preferable "*items*" – a situation that does not exist for summarizers. A survey-based qualitative analysis of the usefulness of model-generated summaries was proposed by Ter Hoeve et al. (2022). Although this work establishes empirically that model-generated summary utility is subjective (as argued in this paper), yet to date, the only work on the formal quantitative evaluation of personalization in summarization models is EGISES (Vansh et al., 2023) which, however, can only capture responsiveness. There is a growing interest in learned summarization evaluation metrics, as opposed to designed ones Peyrard & Eckle-Kohler (2017); Peyrard & Gurevych (2018); Böhm et al. (2019); Liu et al. (2023). LLM-based automated summarization evaluation has also been studied Shen et al. (2023); Zheng et al. (2023a). However, to the best of our knowledge, these works have not focused on evaluating personalized summarization models where subjectivity in saliency needs to be respected (and not normalized). Any such automated evaluator needs to be fine-tuned on large volumes of user profiles having varied subjective expectations. This sort of dataset is hard to design, and such a project only makes sense if designed metrics are not able to achieve minimum human-judgment correlation.

---

[9]In this paper, X: RG-L, RG-SU4, METEOR, BLEU, JSD, BScore (we assume pre-computed embeddings), InfoLM-$\alpha\beta$ which uses BERT Base (uncased; 110M params) as pre-trained Masked Language Model.

[10]System specifications: Machine architecture: $x86\_64$; CPU: Intel(R) Xeon(R) Silver 4216 CPU @ 2.10GHz; CPU Cores: 16; Thread(s) per core: 2.

**Personalized Summarization  Aspect-based.** An aspect-based summarization model generates summaries that are coherent with the aspects (i.e., themes/topics) therein (Narayan et al., 2018; Frermann & Klementiev, 2019; Tan et al., 2020; Hirsch et al., 2021; Hayashi et al., 2021; Meng et al., 2021; Soleimani et al., 2022). While explicit aspects can be restrictive and rather broad (e.g., the MA news training dataset where six broad aspects are identified: "sport", "health", "travel", "news", "science technology", "tv showbiz"), implicit aspect-based summarization implies augmenting the aspect query with concepts that are related to the predefined aspects. Although these models have specific use-cases, they are not trained to adapt to the reader's evolving profile (i.e., reading behavioral pattern) that constitutes discourse-level interest drift (and not just topic-level static interests). Also, the evaluation was w.r.t accuracy using standard ROUGE-variants.

**Interactive Human-feedback-based.** One of the earliest interactive interface-based iterative personalized summarization frameworks was proposed by Yan et al. (2011) where users could click on specific sentences in the generated summary that are of their interest (implicit preference), and read the associated context (i.e., surrounding text) of the selected sentence before sending this preference as feedback to the model for a revised version. A similar interactive interface-based framework was proposed by PVS et al. (2018) where users can iteratively select sentences and the phrases therein that they prefer (and do not prefer as well), while the summarizer, an ILP-based model proposed by Boudin et al. (2015), updates the summary based on this feedback until the user is satisfied. On similar lines, Ghodratnama et al. (2021) introduced a personalized summarization method for extractive summarization. Extracted summary concepts are presented to readers for their feedback, which is used to iteratively fine-tune the summary until no further negative feedback is received. Bohn & Ling (2021) proposed a framework where the acceptance or rejection of summary sentences was made dynamic as the summary gets generated on-the-fly. However, all these works have been evaluated based on standard accuracy evaluations (such as ROUGE variants).

**User-preference Trained Reward Model-based.** Another way of inducing personalization in models is to train a base model (pre-trained or supervised fine-tuning) as an agent within a reinforcement learning framework (usually policy-gradient based) using a reward model (RM) as the environment. The RM is trained on human preferences to predict human ratings for specific actions of the agent (i.e., selecting specific words/sentences), thereby providing the rewards (Stiennon et al., 2020; Nguyen et al., 2022). However, whether these models can explicitly "remember" individual preferences (or even that of user groups with similar interests) is still to be probed and not quite clear. Nevertheless, the evaluations were on accuracy only.

**User-preference-history-based.** There has been considerable work in user preference-history-based product review summarization. The user profile is usually modeled in terms of *discrete attributes* such as rating, user-ID, and product-ID, and *history text* that represents user-written historical review-summaries. Some of the proposed models use user-specific vocabularies to predict *user-preference words* that in turn serve as a guide for generating summaries Ma et al. (2018); Li et al. (2019); Chan et al. (2020). Others have proposed models that learn user preference by jointly training these discrete attributes and the historical review-summaries Liu & Wan (2019); Xu et al. (2021; 2023), where the historical summaries provide information about the writing-style, purchasing preferences, and aspect-of-interest of the users. These works, however, do not fit into the general setup of personalized summarization because the summaries generated are not aligned to a prospective buyer's (i.e., a consumer of review-summary or reader in our parlance) preference behavior, but rather tuned to a *different set of buyers* who are active reviewers and who have provided gold-reference review summaries of their own reviews (i.e., the review-to-summary is a one-to-one mapping and *not subjective*). Nevertheless, the evaluation has been done using accuracy measures (ROUGE variants) only. So far, the only pertinent work incorporating the reader's history as preference is the proposed models that were designed using the PENS framework (Ao et al., 2021), which we studied extensively (see Section 6.2). It is clear from our study that they need significant improvement in terms of personalization (as measured by `PerSEval`).

## 10  Conclusion

In this paper, we presented `PerSEval`, a corrective measure for EGISES (proposed by Vansh et al. (2023)), which, to the best of our knowledge, is the only known personalization measure for summarizers. We first introduced the concept of *responsiveness*, in contrast to *personalization*, as a measure to evaluate the capacity of a model to discern the differences in reader profiles (i.e., reading histories) and generate

reader-specific summaries that maintain this difference proportionately. We then showed that EGISES measures the former. We thereby proved theoretically and empirically on the real-world PENS dataset that measuring responsiveness does not imply measuring personalization since there can be models that generate distinctly different summaries for different reader profiles (i.e., high responsiveness) but are quite off from the expected summaries (i.e., low accuracy and thereby low user-experience (UX)). We then formulated `PerSEval` (more specifically, `DEGRESS`) as a discounted EGISES where the discount factor is a penalty due to accuracy drop called `EDP` (Effective `DEGRESS` Penalty Factor). We analyzed the ten SOTA summarization models using seven variants of `PerSEval` and observed that the model leaderboard reliability depends on the chosen variant. We further observed that the variant `PerSEval`-InfoLM-$\alpha\beta$ performs best regarding rank-stability, a meta-evaluation measure we proposed in this paper. We also proposed a novel survey-based meta-evaluation protocol for human-judgment (HJ) to analyze the extent to which human annotators agree with the design principles of `PerSEval` at a cognitive level. We found that `PerSEval`-InfoLM-$\alpha\beta$ has the highest overall HJ-correlation (Pearson's $r = 0.79$; Spearman's $\rho = 0.68$; Kendall's $\tau = 0.47$). We finally established that separate accuracy leaderboards for personalized summarizers can be misleading and `PerSEval` can serve as a unified measure, thereby emphasizing that personalization and accuracy are inseparable aspects of UX.

## Discussions & Future Directions

**User guideline:** `PerSEval` as a measure is useful in summarization tasks where the user experience of the utility of the summaries generated by any model will be subjective and will depend upon the user's current preferences (which itself is a reflection of past activities). For example, executive summaries and Minutes-of-meetings (MoMs) need to be personalized (as different participants might have different takeaways/todos from the meeting summary) without compromising on the accuracy of the proceedings. In the same way, news recommendation platforms are also likely to benefit from personalized summarization as it increases the likelihood of the article being clicked and read in full if the title or summary is more personalized as per the user profile. In such use cases, `PerSEval` should be adopted rather than accuracy measures or P-Accuracy. It is important to note that _we do not rule out_ _existing conventional accuracy measures and also P-Accuracy_ (as proposed in Vansh et al. (2023)) in favor of `PerSEval`. These measures have their own important role to play for different use cases. Specifically, summarization tasks that do not need personalization should be better evaluated using standard accuracy measures. An example would be the research paper summarization task where saliency, generally, should imply the core contributions of the paper, and hence, the user experience of such summary utility does not depend on specific user preference history. Likewise, there are use cases where P-Accuracy can be more useful, especially when accuracy is the primary objective, yet summarization models to be evaluated also claim to be personalized. An example would be the task of multi-document update summarization for very specific like-minded stakeholders (say, members of a government event-monitoring unit engaging in daily tweets) whose topics of preferences are already known. In this case, P-Accuracy will factor in the lack of personalization (if any) and provide a more realistic accuracy score (and, thereby, the accuracy leaderboard).

**Future directions:** In this work, we analyze the effect of seven variants of `PerSEval` on SOTA summarization models, out of which the ROUGE-variants, BLEU, and METEOR are defined on the string space, JSD is defined on the probability space, BERTScore is defined on the embedding space, and InfoLM is defined on the probability space that is generated from the embedding space using a masked-LM. Although these cover all the most common algebraic spaces on which `PerSEval` can be defined, it remains to be understood how other alternate measures on the same spaces, such as BaryScore (Colombo et al., 2021), MoversScore (Zhao et al., 2019), DepthScore (Staerman et al., 2022), and other variants of InfoLM using multiple Csiszar f-divergences, will behave w.r.t HJ-correlation. Finally, we have only explored one method of estimating `PerSEval`-HJ (i.e., mimicking the human way of computing `PerSEval` using RG-L as the distance between model-generated summary and human-reference) for analyzing HJ-correlations. However, there can be other alternative methods of estimating `PerSEval`-HJ, including incorporating inter-annotator-agreement statistics (such as Kappa statistic). We also consider it a promising future direction to study the effectiveness of `PerSEval` as an objective function to steer the fine-tuning of the personalized summarization models under a Reward Model-driven reinforcement learning setup. In general, the current work can potentially open up directions of systematic evaluation studies of personalization capabilities in models, particularly LLMs with

their In-context Learning capabilities, that apply to all AI challenges where the quality of the model output is subjective to the user's past preferences. We, therefore, encourage the research community to critically study our work and build on it.

## Ethics Statement

We would like to declare that we used the PENS dataset prepared and released by Microsoft Research. Our human-judgment survey was conducted according to the norms set by the Institutional Review Board (IRB) and respects participant anonymity as per guidelines.

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

## A  Model Details

We briefly introduce the SOTA summarization models that were analyzed to understand their degree-of-personalization below:

1. **PENS-NRMS Injection-Type 1**: The PENS framework (Ao et al., 2021) takes user embedding as input along with the news article to generate a personalized summary for that user. To generate user embedding, NRMS (Neural News Recommendation with Multi-Head Self-Attention) (Wu et al., 2019b) is used. It includes a news encoder that utilizes multi-head self-attentions to understand news titles. The user encoder learns user representations based on their browsing history and uses multi-head self-attention to capture connections between news articles. Additive attention is added to learning the news and user representations more effectively by selecting important words and articles. Here, Injection-Type 1 indicates that NRMS user embedding is injected into PENS by initializing the decoder's hidden state of the headline generator, which will influence the summary generation.

2. **PENS-NRMS Injection-Type 2**: To generate a personalized summary, NRMS user embedding is injected into attention values (Injection-Type 2) of PENS that helps to personalize attentive values of words in the news body.

3. **PENS-NAML Injection-Type 1**: NAML (Neural News Recommendation with Attentive Multi-View Learning) Wu et al. (2019a) incorporates a news encoder that utilizes a multi-view (i.e., titles, bodies, and topic categories) attention model to generate comprehensive news representations. The user encoder is designed to learn user representations based on their interactions with browsed news. It also allows the selection of highly informative news during the user representation learning process. This user embedding is injected into the PENS model using Type-1 for personalization.

4. **PENS-EBNR Injection-Type 1**: EBNR (Embedding-based News Recommendation for Millions of Users) Okura et al. (2017) proposes a method for user representations by using an RNN model that takes browsing histories as input sequences. This user embedding is injected using Type 1 into the PENS model for personalization.

5. **PENS-EBNR Injection-Type 2**: This personalized model injects EBNR user embedding into PENS using type-2.

6. **BRIO**: Instead of a traditional MLE-based training approach, BRIO Liu et al. (2022) assumes a non-deterministic training paradigm that assigns probability mass to different candidate summaries according to their quality, thereby helping it to better distinguish between high-quality and low-quality summaries.

7. **SimCLS**: SimCLS (A Simple Framework for Contrastive Learning of Abstractive Summarization) Liu & Liu (2021) uses a two-stage training procedure. In the first stage, a Seq2Seq model (BART (Lewis et al., 2020)) is trained to generate candidate summaries with MLE loss. Next, the evaluation model, initiated with RoBERTa is trained to rank the generated candidates with contrastive learning.

8. **BigBird-Pegasus**: BigBird Zaheer et al. (2020) is an extension of Transformer based models designed specifically for processing longer sequences. It utilizes sparse attention, global attention, and random attention mechanisms to approximate full attention. This enables BigBird to handle longer contexts more efficiently and, therefore, can be suitable for summarization.

9. **ProphetNet**: ProphetNet Qi et al. (2020) is a sequence-to-sequence pre-trained model that employs n-gram prediction using the n-stream self-attention mechanism. ProphetNet optimizes n-step ahead prediction by simultaneously predicting the next n tokens based on previous context tokens, thus preventing overfitting on local correlations.

10. **T5**: T5 (Text-To-Text Transfer Transformer) is based on the Transformer-based Encoder-Decoder architecture that operates on the principle of the unified text-to-text task for any NLP problem, including summarization. Some recent analyses on the performance of T5 on summarization tasks can be found in Tawmo et al. (2022); Ramesh et al. (2022); Etemad et al. (2021).

## B  Accuracy and Performance

### B.1  Accuracy Measures Compared

1. **RG-L**: ROUGE-L (Recall-Oriented Understudy for Gisting Evaluation) (Lin & Och, 2004) calculates the longest common subsequence between the generated summary and the reference summary and then measures the precision, recall, and F1 score based on this comparison.

2. **RG-SU4**: ROUGE-SU4 (Lin, 2004) was designed to consider skip-bigram matches as well, which allows for non-contiguous n-gram matches.

3. **BLEU**: BLEU (Bilingual Evaluation Understudy) (Papineni et al., 2002) is a popular evaluation metric that measures the precision of n-gram matches between the model-generated summaries and the reference summaries. BLEU computes a modified precision score for various n-gram lengths and then combines them using a geometric mean.

4. **METEOR**: METEOR (Metric for Evaluation of Translation with Explicit ORdering) (Banerjee & Lavie, 2005) matches unigrams based on surface forms, stemmed forms, and meanings and then calculates score using a combination of precision, recall, and the order-alignment of the matched words w.r.t reference summary.

5. **Jensen-Shannon Distance**: The Jensen-Shannon Distance (JSD) (Menéndez et al., 1997) is a metric used in summarization evaluation to measure the dissimilarity between probability distributions of words in a reference summary and a generated summary. It quantifies the information divergence and similarity, providing a nuanced assessment of the semantic content overlap between the two summaries.

6. **BertScore**: BertScore (BScore) (Zhang et al., 2020) is a metric for evaluating machine-generated summaries, emphasizing contextual embeddings from BERT to assess both word overlap and contextual relationships. It overcomes the limitations of keyphrase-based measures like ROUGE.

7. **InfoLM-$\alpha\beta$**: Given a user-generated reference summary $u$ and a model-generated summary $s_u$, InfoLM (Colombo et al., 2022b) recursively masks each token position $k$ of both $u$ (denoted $[\boldsymbol{u}]^k$) and $s_u$ (denoted $[\boldsymbol{s_u}]^k$) to obtain individual masked contexts of length $l_u$ and $l_{s_u}$ respectively. For each masked context, it uses a pre-trained masked-language model to estimate the corresponding probability distribution over the vocabulary (i.e., $p_\theta \left( \cdot \mid [\cdot]^k ; M_{\boldsymbol{\theta},h} \right)$), resulting in two bags of distributions of size $l_u$ and $l_{s_u}$ for $u$ and $s_u$. The bags of distributions (for both masked $u$ and masked $s_u$) are then averaged out, as follows[11]:

$$p(\cdot \mid \boldsymbol{u}; M_{\boldsymbol{\theta},h}) \triangleq \sum_{k=1}^{l_u} \gamma_k \times p_\theta \left( \cdot \mid [\boldsymbol{u}]^k ; M_{\boldsymbol{\theta},h} \right)$$

$$p(\cdot \mid \boldsymbol{s_u}; M_{\boldsymbol{\theta},h}) \triangleq \sum_{k=1}^{l_{su}} \gamma_k \times p_\theta \left( \cdot \mid [\boldsymbol{s_u}]^k ; M_{\boldsymbol{\theta},h} \right)$$

InfoLM then uses a chosen information measure $\mathcal{I}$ to compute the following:

$$\text{InfoLM}(\mathbf{u}, \mathbf{s_u}) \triangleq \mathcal{I}\left[ p(\cdot \mid \boldsymbol{u}), p(\cdot \mid \mathbf{s_u}) \right]$$

In our experiments, we chose $\mathcal{I}$ to be $\alpha\beta$-divergence (also called AB-Divergence; $\mathcal{D}_{AB}^{\alpha,\beta}$) (Cichocki et al., 2011) where the divergence is defined as:

$$\mathcal{D}_{AB}^{\alpha,\beta} = \frac{1}{\beta(\beta+\alpha)} \log \sum p_i^{\beta+\alpha} + \frac{1}{\beta+\alpha} \log \sum q_i^{\beta+\alpha} - \frac{1}{\beta} \log \sum p_i^\alpha q_i^\beta \tag{8}$$

## B.2  Model Rank Aggregation & Agreement

1. **Borda-Kendall Consensus based Rank Aggregation:** The Borda-Kendall (BK) consensus entails aggregating a set of permutations, denoted as $\eta^1, \ldots, \eta^L \in \mathfrak{N}$, which represent the rankings of $N$ models across $L \geq 1$ tasks or instances (in our case, the pair of accuracy rank measure and the PerSEval-variant to be aggregated). This aggregation involves summing the ranks of each model and subsequently ranking the obtained sums. Formally:

$$\text{sum}_n := \sum_{l=1}^{L} \eta_n^l \text{ for every } 1 \leq n \leq N,$$

$$\text{BK}\left( \eta^1, \ldots, \eta^L \right) = \text{argsort}\left( \text{sum}_1, \ldots, \text{sum}_T \right)$$

2. **Pearson's Correlation Coefficient** $(r)$:

$$r = \frac{\sum_{i=1}^n (x_i - \overline{x})(y_i - \overline{y})}{\sqrt{\sum_{i=1}^n (x_i - \overline{x})^2 \sum_{i=1}^n (y_i - \overline{y})^2}}$$

where $\overline{x}, \overline{y}$ are the means of the variables $x_i$ and $y_i$ ; $n =$ the number of samples.

3. **Spearman's $\rho$ Coefficient**:

$$\rho = 1 - \frac{6 \sum d_i^2}{n(n^2 - 1)}$$

where $d =$ the pairwise distances of the ranks of the variables $x_i$ and $y_i$ ; $n =$ the number of samples.

4. **Kendall's $\tau$ Coefficient**:

$$\tau = \frac{c - d}{c + d} = \frac{S}{\binom{n}{2}} = \frac{2S}{n(n-1)}$$

where, $c =$ the number of concordant pairs; $d =$ the number of discordant pairs.

---

[11] $\gamma_k$ are measures of the importance of the $k$-th token in $u$ and $s_u$, respectively s.t. $\sum_{k=1}^{l_u} \gamma_k = \sum_{k=1}^{l_{su}} \gamma_k = 1$. $\gamma_k$ are computed using the corpus-level inverse document frequency (IDF) scores.

| PENS Test Dataset Sample Set (Random Selection) | | | | | | | |
|---|---|---|---|---|---|---|---|
| $\delta$-Bias (in **bold**) of PSE-variants | | | | | | | |
| **Models** | **PSE-RG-L** | **PSE-RG-SU4** | **PSE-METEOR** | **PSE-BLEU** | **PSE-JSD** | **PSE-BScore** | **PSE-InfoLM-$\alpha\beta$** |
| BigBird-Pegasus | 0.0009 | 0.0034 | 0.001 | 0.0031 | 0.0011 | 0.0015 | **0.0024** |
| SimCLS | 0.0034 | 0.007 | 0.0049 | **0.0054** | **0.0019** | **0.0017** | 0.0018 |
| BRIO | **0.0041** | **0.0072** | **0.0055** | 0.0051 | 0.001 | **0.0017** | 0.0013 |
| ProphetNet | 0.0033 | 0.0059 | 0.0041 | 0.0052 | 0.0002 | 0.0015 | 0.0017 |
| T5 (Base) | 0.0035 | 0.0049 | 0.0047 | 0.0032 | 0.0004 | 0.0016 | 0.0012 |
| PENS-NAML T1 | 0.0033 | 0.0011 | 0.0027 | 0.0003 | 0.0001 | 0.0016 | 0.0012 |
| PENS-NRMS T1 | 0.0029 | 0.0003 | 0.0033 | 0.0008 | 0.0003 | 0.0016 | 0.0006 |
| PENS-EBNR T1 | 0.0035 | 0.0004 | 0.0039 | 0.0002 | 0.0001 | 0.0016 | 0.0008 |
| PENS-NRMS T2 | 0.0038 | 0.0002 | 0.0042 | 0.0003 | 0.0001 | 0.0016 | 0.0003 |
| PENS-EBNR T2 | 0.0036 | 0.0007 | 0.0042 | 0.0002 | 0 | 0.0015 | 0.0004 |
| $\delta$-Variance (in **bold**) of PSE-variants | | | | | | | |
| **Models** | **PSE-RG-L** | **PSE-RG-SU4** | **PSE-METEOR** | **PSE-BLEU** | **PSE-JSD** | **PSE-BScore** | **PSE-InfoLM-$\alpha\beta$** |
| BigBird-Pegasus | 7.74E-07 | 1.17E-05 | 1.05E-06 | 9.43E-06 | 1.14E-06 | 2.36E-06 | **5.81E-06** |
| SimCLS | 1.14E-05 | 4.86E-05 | 2.44E-05 | **2.92E-05** | **3.64E-06** | **2.82E-06** | 3.08E-06 |
| BRIO | **1.65E-05** | **5.25E-05** | **3.07E-05** | 2.61E-05 | 9.86E-07 | 2.75E-06 | 1.77E-06 |
| ProphetNet | 1.09E-05 | 3.47E-05 | 1.69E-05 | 2.71E-05 | 3.20E-08 | 2.39E-06 | 2.90E-06 |
| T5 (Base) | 1.21E-05 | 2.43E-05 | 2.17E-05 | 9.99E-06 | 1.54E-07 | 2.61E-06 | 1.33E-06 |
| PENS-NAML T1 | 1.12E-05 | 1.13E-06 | 7.51E-06 | 8.96E-08 | 4.00E-09 | 2.43E-06 | 1.41E-06 |
| PENS-NRMS T1 | 8.65E-06 | 9.84E-08 | 1.09E-05 | 6.14E-07 | 9.44E-08 | 2.43E-06 | 3.26E-07 |
| PENS-EBNR T1 | 1.22E-05 | 1.58E-07 | 1.53E-05 | 5.20E-08 | 6.40E-09 | 2.53E-06 | 6.00E-07 |
| PENS-NRMS T2 | 1.41E-05 | 5.44E-08 | 1.79E-05 | 1.02E-07 | 1.36E-08 | 2.41E-06 | 1.14E-07 |
| PENS-EBNR T2 | 1.30E-05 | 4.94E-07 | 1.75E-05 | 3.76E-08 | 0 | 2.31E-06 | 1.84E-07 |
| Summary | | | | | | | |
| **Models** | **PSE-RG-L** | **PSE-RG-SU4** | **PSE-METEOR** | **PSE-BLEU** | **PSE-JSD** | **PSE-BScore** | **PSE-InfoLM-$\alpha\beta$** |
| $\delta$-stability | 0.0041 | **0.0072** | 0.0055 | 0.0054 | 0.0019 | 0.0017 | 0.0024 |
| $\epsilon$-Spearman | 1 | 1 | 1 | 1 | 1 | 1 | 1 |
| $\epsilon$-Kendall | 1 | 1 | 1 | 1 | 1 | 1 | 1 |

Table 6: **PerSEval**$^{\beta=1.7}$ **Stability**: 0.0072-strongly-stable w.r.t $\epsilon$-Spearman = 1; $\epsilon$-Kendall = 1 across variants

# C    Detailed Experimental Results

We provide a detailed analysis of all the seven PerSEval variants in terms of their stability performance in the following section.

## C.1    PerSEval Stability Results

In this section, we provide a detailed analysis of the stability performance of all the seven PerSEval variants (in Section 8.1, we discussed that of the best performing PerSEval-InfoLM variant only). We analyze the $\delta$-bias and the $\delta$-variance of each variant across all the ten SOTA models that have been studied. We observe that while the best-performing variant w.r.t bias is PerSEval-BertScore and w.r.t variance is PerSEval-RG-L, the worst performances w.r.t both are pretty low with an overall 0.0072 $\delta$-stability across all the variants (see Table 6). We also observed a consistent 100% rank-correlation (i.e., $\epsilon$-stability) across all the variants, showing PerSEvalto be extremely stable.

## C.2    EGISES and P-Acc Results

| Models | EG-RG-L | EG-RG-SU4 | EG-METEOR | EG-BLEU | EG-JSD | EG-BScore | EG-InfoLM-$\alpha\beta$ |
|---|---|---|---|---|---|---|---|
| BigBird-Pegasus | 0.324 | 0.097 | 0.356 | 0.128 | 0.387 | 0.390 | 0.33 |
| SimCLS | 0.429 | 0.268 | 0.501 | 0.376 | 0.512 | 0.472 | 0.28 |
| BRIO | 0.480 | 0.282 | 0.581 | 0.434 | 0.630 | 0.572 | 0.189 |
| ProphetNet | 0.548 | 0.431 | 0.602 | 0.501 | 0.608 | 0.577 | 0.289 |
| T5 (Base) | 0.592 | 0.474 | 0.642 | 0.551 | 0.641 | 0.619 | 0.225 |
| PENS-NAML T1 | 0.869 | 0.702 | 0.866 | 0.798 | 0.883 | 0.863 | 0.575 |
| PENS-NRMS T1 | 0.891 | 0.745 | 0.887 | 0.832 | 0.901 | 0.886 | 0.654 |
| PENS-EBNR T1 | 0.935 | 0.820 | 0.931 | 0.892 | 0.938 | 0.933 | 0.793 |
| PENS-EBNR T2 | 0.986 | 0.879 | 0.983 | 0.958 | 0.981 | 0.986 | 0.907 |
| PENS-NRMS T2 | 0.988 | 0.908 | 0.986 | 0.965 | 0.983 | 0.989 | 0.824 |

Table 7: SOTA Responsiveness Benchmarking on Microsoft PENS Dataset w.r.t seven EGISES (EG) variants.

| Models | P-Acc-RG-L | P-Acc-RG-SU4 | P-Acc-METEOR | P-Acc-BLEU | P-Acc-JSD | P-Acc-BScore | P-Acc-InfoLM-$\alpha\beta$ |
|---|---|---|---|---|---|---|---|
| T5 (Base) | 0.492 | 0.591 | 0.532 | 0.594 | 0.411 | 0.123 | 0.392 |
| ProphetNet | 0.481 | 0.580 | 0.524 | 0.588 | 0.393 | 0.121 | 0.368 |
| BRIO | 0.468 | 0.582 | 0.520 | 0.593 | 0.333 | 0.113 | 0.344 |
| SimCLS | 0.446 | 0.558 | 0.500 | 0.577 | 0.340 | 0.107 | 0.301 |
| BigBird-Pegasus | 0.412 | 0.517 | 0.438 | 0.508 | 0.304 | 0.119 | 0.382 |
| PENS-NAML T1 | 0.391 | 0.523 | 0.404 | 0.518 | 0.350 | 0.106 | 0.522 |
| PENS-NRMS T1 | 0.385 | 0.518 | 0.398 | 0.512 | 0.344 | 0.104 | 0.510 |
| PENS-EBNR T1 | 0.383 | 0.514 | 0.398 | 0.508 | 0.341 | 0.103 | 0.512 |
| PENS-NRMS T2 | 0.379 | 0.507 | 0.394 | 0.501 | 0.338 | 0.101 | 0.492 |
| PENS-EBNR T2 | 0.376 | 0.506 | 0.390 | 0.500 | 0.336 | 0.102 | 0.504 |

Table 8: SOTA Personalized Accuracy Benchmarking on Microsoft PENS Dataset w.r.t P-Accuracy (P-Acc); P-Acc hyper-parameters: $\alpha = 0.5$, $\beta = 1$ (i.e., complete importance to lack of personalization).

# D    Survey Format: Human-Judgment Meta-evaluation of `PerSEval`

In this section, we present the screenshot of the questionnaire designed for the survey for computing the human-judgment version of `PerSEval` (`PerSEval`-HJ). Two consecutive respondents evaluated the generated summary pairs of all ten benchmarked models.

## Evaluation Metric Correlation Survey

You are supposed to rate the sentence pair based on *similarity*.
**The meaning of each score is given below.**
1: Almost different, 2: Very dissimilar, 3: Somewhat dissimilar, 4: Somewhat similar, 5: Very similar, 6: Almost same

Your Name (optional)

Your gender:

○ Male ○ Female ○ Transgender ○ Prefer not to say

Your occupation:

○ Undergrad student ○ Grad student ○ Teacher ○ Corporate Professional ○ Other

*Sentence 1:* gary woodland drained a 50-foot birdie put at his final hole on friday to cap a six-under par 65

*Sentence 2:* gary woodland drained a 50-foot birdie put at his final hole on friday to cap a six-under par 65 and take a two-stroke us open lead over former champion justin rose at pebble beach.

○ 1 ○ 2 ○ 3 ○ 4 ○ 5 ○ 6

*Sentence 1:* gary woodland drained a 50-foot birdie put at his final hole on friday to cap a six-under 65. woodland's tee shot at his final hole, the par-four ninth, nestled in a divot in the fairway. woodland becomes just the third player to post a 65 in us open play at pebble beach. his 36-hole total of nine-under 133 is one shot better than woods posted in 2000.

*Sentence 2:* gary woodland drained a 50-foot birdie put at his final hole on friday to cap a six-under 65. woodland's tee shot at his final hole, the par-four ninth, nestled in a divot in the fairway. woodland becomes just the third player to post a 65 in us open play at pebble beach.

○ 1 ○ 2 ○ 3 ○ 4 ○ 5 ○ 6

*Sentence 1:* gary woodland shoots six-under par 65 to take two-stroke us open lead over former champion justin rose at pebble beach. woodland drains 50-foot birdie put at final hole to cap round of 65. woodland becomes just the third player to post a 65 in us open play at the course.

*Sentence 2:* gary woodland shoots six-under par 65 to take two-stroke us open lead at pebble beach. former champion justin rose two shots back after second round of par-par 65. woodland becomes just the third player to post a 65 in us open play at the beach.

○ 1 ○ 2 ○ 3 ○ 4 ○ 5 ○ 6

*Sentence 1:* gary woodland drains a 50-foot birdie put at his final hole on friday to cap a six-under par 65. woodland takes a two-stroke us open lead over former champion justin rose at pebble beach. four-time major winner rory m

*Sentence 2:* gary woodland drained a 50-foot birdie put at his final hole to cap a six-under par 65 and take a two-stroke us open lead over former champion justin rose. woodland became just the third player to post a 65 in us open play at pebble

○ 1 ○ 2 ○ 3 ○ 4 ○ 5 ○ 6

*Sentence 1:* gary woodland drained a 50 - foot birdie put at his final hole to cap a six - under par 65 at pebble beach. former champion justin rose fires a second - round 70 for 137 to take a two - stroke lead. four - time major winner rory mcilroy and south african - born american aaron wise are a shot back on five - under 137.

*Sentence 2:* gary woodland drained a 50 - foot birdie put at his final hole to cap a six - under par 65. woodland takes a two - stroke lead over former champion justin rose at pebble beach. four - time major winner rory mcilroy and south african - born american aaron wise are a shot back on five - under 137.

○ 1 ○ 2 ○ 3 ○ 4 ○ 5 ○ 6

*Sentence 1:* A pair of players are tied at four-under, a group that included two-time defending us open champ justin rose at pebble beach. woodlands tee shot at his final hole, the par-four ninth, nestled in a divot in the fairway, but he still managed to reach the green in two to close out his round in sensational style.

*Sentence 2:*  A two-stroke us open lead over former champion justin rose at pebble beach. A two-stroke us open lead over former champion justin rose at pebble beach. A three-time winner on the pga tour who led last year us open championship at the halfway stage on the way to his best major finish -- a tie for sixth-- has tee shot at his final hole, the par-four ninth, nestled in a divot in the fairway, but he still managed to reach the green in two to close out his round in sensational style.

○ 1 ○ 2 ○ 3 ○ 4 ○ 5 ○ 6

**SUBMIT**

Disclaimer: This data is solely for research purpose. You may optionally add your name, which will be added to our contributor list when this dataset will be published.

Figure 4: **Sample Questionnaire**: Six pairs of summaries for a specific document; five pairs are model-generated summaries (each user evaluates five of the ten models) for a specific document, while one pair is user-generated gold reference).

