Dear Action Editors and Reviewers,

We thank the reviewers for their time and valuable feedback. We also deeply appreciate your prompt action to include a fourth reviewer without delaying the final conclusion on the acceptance of the work any further. In this response letter, along with the suggestion and response to the new reviewer j1ks, we have also included the previous reviewers' suggestions (together with our responses). We use italics for the reviewer comments, black color for our responses, and blue color for excerpts of what we have added to the revised manuscript. As before, we have also appended a color-coded copy of the revised manuscript here with the new additions in red along with the previous additions that are marked in blue.

# 1 Review of Paper2691 by Reviewer hE9R

**Summary of Suggested Revisions:**

1. *Clarification regarding the broader scope of the work. –*

   **Response:** To address the limitation being restricted to the Microsoft PENS dataset, we have proposed **a novel appropriation methodology of approximately simulating personalization evaluation setup using gold-standard datasets**. We used the OpenAI CNN/Daily-Mail and OpenAI TL;DR (Reddit) datasets that consist of RLHF-tuned (PPO-based) summarization policies and corresponding human feedback (i.e., ratings). We generated two additional baselines on these datasets. More details can be found in sections 6.1, 6.2, and Table 1.

# 2 Review of Paper2691 by Reviewer bX5T

**Summary of Suggested Revisions:**

1. *Regarding discussion on P-Acc and its difference with EGISES and the possibility of it being an alternative to* `PerSEval` *–*

   **Response:** – The discussion has been included in the Introduction and section 4: Limitations of P-Accuracy (`P-Acc`) as a Solution to the EGISES Paradox). In this section, we proved three limiting properties of P-Acc, showing why the measure can be used for regulating the accuracy of models but cannot be directly used for measuring personalization. We also provided detailed empirical P-Acc results (Table 8) on the studied models, showing that even though models have poor EGISES scores, they fare closely with models having significantly better EGISES scores, thereby showing that, as originally

intended by Vansh et al. (2023), the measure is primarily dominated by the accuracy score. Since we established that P-Acc is not an alternative measure, we do not see the need for doing human-judgment correlation in the context of personalization evaluation as **PerSEval and P-Acc are not comparable.**

2. *Regarding applicability and comparability of Human-judgment based Correlation Meta-evaluation –*

   **Response:** – We do not need such comparisons since (a) P-Acc is not an alternative measure, and (b) EGISES is an incomplete measure with respect to personalization evaluation (as established both formally in section 3 and empirically in section 8.1 and in Table 3).

3. *Clarification on the rationale for better performance of PerSEval-InfoLM-$\alpha\beta$ –*

   **Response:** This has been provided in section 8.1: Reliability of `PerSEval`.

4. *Regarding discussion on the comparison of `PerSEval`(or designed metrics) with learned metrics –*

   **Response:** This has been added in section 9: Evaluation of Personalization.

5. *Clarification with better example on the trade-off between personalization and accuracy -*

   **Response:** This has been explained in detail with a rewriting of section 2.1.

## 3   Review of Paper2691 by Reviewer muUC

**Summary of Suggested Revisions:**

1. *Clarification regarding examples in page 3. –*

   **Response:** This has been explained in detail with a rewriting of section 2.1.

2. *Regarding reliability and utility w.r.t other datasets. –*

   **Response:** To address the limitation being restricted to the Microsoft PENS dataset, we have proposed **a novel appropriation methodology of approximately simulating personalization evaluation setup using gold-standard datasets**. We used the OpenAI CNN/Daily-Mail and OpenAI TL;DR (Reddit) datasets that consist of RLHF-tuned (PPO-based) summarization policies and corresponding human feedback (i.e., ratings). We generated two additional baselines on these datasets. More details can be found in sections 6.1, 6.2, and Table 1.

3. *Regarding computational implication of* `PerSEval` *–*

    **Response:** This has been provided in section 8.3.

# 4  Review of Paper2691 by Reviewer j1ks

**Summary of Suggested Revisions:**

1. *Regarding addition of user guideline explaining the use-cases of* `PerSEval` *–*

    **Response:** Thank you for pointing this out. We also felt this is much needed to put the overall contribution in the right perspective. We have added a whole paragraph on this in the Discussion and Future Work section (marked in red).

# 5  TYPO

Please note that there was a typo in the explanation of bounding cases of Theorem 4 in section 4 (page 7) of the earlier revision. We have rectified it in this revised manuscript and have marked it in red in the appended copy.

# References