# OpenReview forum: "PerSEval: Assessing Personalization in Text Summarizers"
_TMLR — Accepted by TMLR_

### Review · Reviewer_muUC · 2024-06-23

**Summary Of Contributions:**

This paper introduces **PerSEval**, a **metric** for evaluating the **personalization of text summarization** models. The authors find that the most popular personalization evaluation metric --- **EGISES** --- only measures the degree of responsiveness instead of the actual personalization. To better eval the personalization, the authors propose PerSEval. It adds a penalty for accuracy drop into EGISES to create a more personalized evaluation. The authors test PerSEval using ten state-of-the-art summarization models and find it to be very reliable, with a high correlation with human judgment and high rank stability. They also demonstrate that relying only on accuracy-based aggregated results can lead to misleading conclusions. The authors argue that the PerSEval offers a more comprehensive evaluation of personalization in text summarizers, potentially improving user experience.

**Audience:**

Yes

**Broader Impact Concerns:**

1.  the use of personal user data

**Claims And Evidence:**

Yes

**Requested Changes:**

1. The examples in page 3 (Alice) is not clear and confusing.

2. Other datasets results are needed to prove reliability. I am not convinced by its usefulness.

3. Discuss the computational implications of using PerSEval, such as the additional resources required due to the complexity of the method.

**Strengths And Weaknesses:**

Strengths:

1.The authors find a gap in the evaluation of text personalization models. The PerSEval tries to distinguish between responsiveness and personalization in this context. It looks like a good new metric in this field.

2. PerSEval incorporates an **accuracy penalty** into the evaluation. lt offers a more comprehensive measure of how well a model personalizes text.

3. The authors how an thorough empirical analysis with a range of state-of-the-art models. It shows that this new method has a high reliability and rank stability.

4. I believe this paper and metric potentially influences future research in the field.

Weaknesses:

1. Too few dataset: The analysis only uses PENS dataset. I am not convinced that it will also work on other datasets.

2. The paper doesn't explore how alternative measures to those chosen for empirical testing perform regarding the human-judgement correlation for PerSEval.

3. How to train model which can optimize for PerSEval, given its complexity?

---

> ### Author Response · Authors · 2024-07-25
> **Response to the suggestions by Reviewer muUC**
>
> We thank the reviewers for their time and valuable feedback. We also deeply appreciate your patience and consideration, given the medical emergency that we had to face. We sincerely apologize for the delay. We believe the suggestions have helped us improve the quality of the manuscript.
>
> 1. $\textcolor{red}{\textit{Clarification regarding examples in page 3.}}$:  This has been explained in detail with a rewriting of section 2.1.
>
> 2. $\textcolor{red}{\textit{Regarding reliability and utility w.r.t other datasets}}$:  To address the limitation of being restricted to the Microsoft PENS dataset, we have proposed $\textbf{a novel appropriation methodology of approximately simulating personalization evaluation setup using gold-standard datasets}$. We used the OpenAI CNN/Daily-Mail and OpenAI TL;DR (Reddit) datasets that consist of RLHF-tuned (PPO-based) summarization policies and corresponding human feedback (i.e., ratings). We generated two additional baselines on these datasets. More details can be found in sections 6.1, 6.2, and Table 1.
>
> 3. $\textcolor{red}{\textit{Regarding computational implication of PerSEval}}$: This has been provided in section 8.3.

---

### Review · Reviewer_bX5T · 2024-06-28

**Summary Of Contributions:**

The paper proposes an extension to existing personalized text summarization benchmark EGISES, enriching it with an additional accuracy criterion. The new method is referred to as PerSEval. The authors explain and provide a proof for a shortcoming of EGISES, where a summary can be ranked high in personalization while having low accuracy, resulting in a paradox. The baseline of the argumentation is that EGISES not measures personalization (which would imply high user experience), but a weaker concept referred to as responsiveness. The novel criterion focusses on penalizing low accuracy on top of EGISES, while making sure that high accuracy does not deacrease the personalization. More specifically, the authors add a accuracy drop penalty to ensure sufficiently high accuracy and accuracy-inconsistency penalty to prevent outliers in accuracy scores.

The approach is evaluated for the PENS dataset, following the EGISES baseline paper. The ranking is calculated for the same set of SOTA summarization models. For the evaluation, 7 metrics, including ROUGE (RG)-L, Meteor and BertScore, are used as distance metric for the proposed score. The results show that their variant PerSEval-InfoLM-αβ provides the best discriminative performance.

In addition, a meta evaluation was conducted with 169 students in order to find out if PerSEval comes close to human evaluations. The authors therefore had the participants compare summarized sentences in terms of their similarity. PerSEval-InfoLM-αβ seems to again provide results closest to human judments.

The authors show also show that their approach is not perfectly correlated with EGISES by comparing their outputs and conduct a final evaluation to assess the stability of resulting ranks by sampling different amounts of summaries from PENS. The authors infer that the proposed approach is stable and robust.

**Audience:**

Yes

**Broader Impact Concerns:**

Broader impact concerns are available in the paper. Since the concept personalization vs. responsiveness vs. UX vs. accuracy is somewhat subjective and philosophical, it might be beneficial to add such a discussion to the section.

**Claims And Evidence:**

No

**Requested Changes:**

Based on the possible, perceived weaknesses, I would propose the following changes:
* Please add a distinct discussion of the difference of P-Acc to EGISES. Optimally this should be worked in to the introduction and main argumentation for the paper as well, as this is the main piece of related work here. Does P-Acc potentially not solve the problem and one needs a better metric? Can we still make the triangulation proof on the basis of P-Acc? Is the ranking of P-Acc inferior to PerSEval? Until this is clarified, I would not agree that claims and evidence are clear.
* Would it be able to infer more information from the human judgement experient you conducted which would allow to compare EGISES / P-Acc / PerSEval on further? If so, please add such information or provide argumentation why this not possible / not needed at all.
* Why are other variants of your system not as competing as PerSEval-InfoLM-αβ? Please add more discussions on your results, helping the reader to take informed design decisions.
* Please add a discussion on the boundaies of such a crafted metric to a learned metric, which is not possible to be learned without feedback data. To what extent does the defined personalization property hold?
* The presentation of the paper does not include sufficient examples for the accuracy measure for the claimed difference to personalization. Maybe this is too basic, but I am missing a better example on what accuracy means for a summary - is it false statements, leaving out central parts of the content or both? Could a user not want to trade in accuracy (in terms of removing some parts of the content) for a better focus on a subtopic of interest? Where are the boundaries here, what are the core assumptions? How were they taken?

**Strengths And Weaknesses:**

# Strenghts:
* The argumentation of requiring an accuracy factor next to an accuracy-independent personalization score (or the defined responsiveness) is sensible
* The conducted evaluation is significantly aligned to the one for EGISES, comprising the same SOTA baseline summarizers as well as the same sampling tests for robustness.
* The conducted evaluation comprises a user study with a rather large set of participants, where a sensible approach is chosen to assess the approach's capability to replicate human judement.
* The results concerning the individual performance of PerSEval seem thorough, providing a sensible ranking and showing sufficient correlation to human judgements.

# Weaknesses:
* A main concern I have on the paper is the missing mention and comparison to the combined personalization plus accuracy metric the authors of EGISES suggest at the end of their paper, namely P-Acc. It therefore remains unclear from the paper what the main novelty is.
* The evaluation only consists of one direct comparison to EGISES, where the correlation is investigated. A deeper analysis, also with respect to human judgements is missing. It is unclear which method the participants would prefer.
* Similarly, a deep analysis of Table 2 results, comprising the human judgement evaluation, is missing. Besides mentioning the best performing PerSEval-InfoLM-αβ, no insights are drawn with respect to the used distance metrics which do not perform as well.
* The related work touches on works RecSys metric using human feedback which is ruled of for text sumarizers. I understand that available datasets might be generated in a static, offline manner, prohibiting a thorough evaluation. What is not clear to me are the boundaries of the taken approach when defining personalization on the history of summaries.
* Since human feedback based summarizers can gradually improve a summary, an argumentation is missing why one would use the proposed metric on such approaches within the human judgement meta evaluation.

---

> ### Author Response · Authors · 2024-07-25
> **Response to the suggestions by Reviewer bX5T**
>
> We thank the reviewers for their time and valuable feedback. We also deeply appreciate your patience and consideration, given the medical emergency that we had to face. We sincerely apologize for the delay. We believe the suggestions have helped us improve the quality of the manuscript.
>
> 1. $\textcolor{red}{\textit{Regarding discussion on P-Acc and its difference with EGISES and the possibility of it being an alternative to PerSEval}}$: The discussion has been included in the Introduction and section 4: $\textcolor{blue}{Limitations of P-Accuracy (P-Acc) as a Solution to the EGISES Paradox}$). In this section, we proved three limiting properties of P-Acc, showing why the measure can be used for regulating the accuracy of models but cannot be directly used for measuring personalization. We also provided detailed empirical P-Acc results (Table 8) on the studied models, showing that even though models have poor EGISES scores, they fare closely with models having significantly better EGISES scores, thereby showing that, as originally intended by Vansh et.al, the accuracy score primarily dominates the measure. Since we established that P-Acc is not an alternative measure, we do not see the need for doing human-judgment correlation in the context of personalization evaluation as $\textbf{PerSEval and P-Acc are not comparable.}$
>
> 2. $\textcolor{red}{\textit{Applicability and comparability of Human-judgment based Correlation Meta-evaluation}}$ : We do not need such comparisons since (a) P-Acc is not an alternative measure, and (b) EGISES is an incomplete measure with respect to personalization evaluation (as established both formally in section 3 and empirically in section 8.1 and in Table 3).
>
> 3. $\textcolor{red}{\textit{Rationale for better performance of PerSEval-InfoLM-$\alpha\beta$}}$: This has been provided in section 8.1: Reliability of PerSEval.
>
> 4. $\textcolor{red}{\textit{Comparasion of PerSEval (or designed metrics) with learned metrics}}$: This has been added in section 9: Evaluation of Personalization where we cited several works on learned metrics and also LLM-based evaluation along with the limitations.
>
> 5. $\textcolor{red}{\textit{Clarification with better example on the trade-off between personalization and accuracy}}$: This has been explained in detail with a rewriting of section 2.1.

---

### Review · Reviewer_hE9R · 2024-07-03

**Summary Of Contributions:**

This paper focuses on assessing personalization in text summarizers. Concretely, this paper proposes PerSEval, a corrective measure for EGISES proposed by Vansh et al. (2023). By introducing the concept of responsiveness as opposed to personalization, this paper finds that EGISES only measures the ability of discerning the differences in reader profiles (i.e., reading histories). However, it is also necessary to generate reader-specific summaries that maintain this difference proportionately. One cannot expect high user experience while having low accuracy. This paper then mathematically proves that a model can have a very good EGISES score but may still fall short of being personalized simply because of low accuracy performance. As a result, this work proposes PerSEval that builds on the design principles of EGISES and bridges the gap of user experience. Analysis results show that  PerSEval provides a much more reliable ranking of models with significantly higher human-judgment correlation.

**Audience:**

Yes

**Broader Impact Concerns:**

The broader impact of this paper has not been clearly justified yet.

**Claims And Evidence:**

Yes

**Requested Changes:**

1. Justify the significance of assessing personalization in general text summarization problems.

2. Fix the reference mistakes.

**Strengths And Weaknesses:**

Strengths:

1. It is reasonable to distinguish responsiveness from personalization and then identify the shortcoming of the existing EGISES measure. This is well motivated that there indeed can be scenarios where a model exhibits high responsiveness at the cost of losing accuracy. Under such conditions, the EGISES measure could be risky.

2. This paper theoretically and empirically show that the propose  PerSEval measures the degree of responsiveness, a necessary but not sufficient condition for degree-of-personalization. It makes sense to conclude that the EGISES-paradox not just theoretically exists but has real evidence.

3. The experiments are generally informative and convincing. Two ten SOTA summarization models were assessed. Based on the results, this paper empirically establishes that PerSEval is reliable w.r.t human-judgment correlation (Pearson’s r = 0.73; Spearman’s ρ = 0.62; Kendall’s τ = 0.42). PerSEval has high rank-stability and PerSEval as a rank measure is not entailed by EGISES-based ranking. In addition, PerSEval can be a standalone rank-measure without the need of any aggregated ranking

Weaknesses:

This paper is overall comprehensive and well written. I did not find major concerns. While this paper seems to be generally sound, the research scope is somewhat narrow. As the authors have already argued that they have not found any containing user behavior history, such as user-click timestamp records, other than the news dataset, the generality could be limited. On the one hand, it seems hard to collect the data with user behavior history in reality. On the other hand, this work is a correction for a specific metric. It remains unclear how this work could facilitate related studies and inspire future work. Any thoughts on those aspects would be beneficial.

Minor Comments:

There is a reference mistake in the first paragraph (Ln 6) of Section 1, which shows "?". Besides, the authors may double check the format of the references and use \citet{} and \citep{} correctly.

---

> ### Author Response · Authors · 2024-07-25
> **Response to suggestions made by Reviewer  hE9R**
>
> We thank the reviewers for their time and valuable feedback. We also deeply appreciate your patience and consideration, given the medical emergency that we had to face. We sincerely apologize for the delay. We believe it has helped us improve the quality of the manuscript.
>
> $\textcolor{red}{\textit{Clarification regarding broader scope of the work}}$: To address the limitation being restricted to the Microsoft PENS dataset, we have proposed $\textbf{a novel appropriation methodology of approximately simulating personalization evaluation setup using gold-standard datasets}$. We used the OpenAI CNN/Daily-Mail and OpenAI TL;DR (Reddit) datasets that consist of RLHF-tuned (PPO-based) summarization policies and corresponding human feedback (i.e., ratings). We generated two additional baselines on these datasets. More details can be found in sections 6.1, 6.2, and Table 1.

---

### Review · Reviewer_j1ks · 2024-09-03

**Summary Of Contributions:**

This paper starts by calling out neither traditional accuracies metrics nor EGISES proposed by recent study are sufficient enough to evaluate degree of personalization of summarizers. The authors empirically proved EGISES measures only the degree of responsiveness and illustrated with examples that EGISES scores can stay unchanged when accuracies vary through experiments. To measure both responsiveness and accuracies, the authors introduced a new evaluation metric called PerSEval, which addressed the limitations of both EGISES and Accuracy with the help of a penalty factor called EDP. The authors also empirically proves that PerSEval shows high correlations to human judgement comparing to other metrics through multiple robust experiments.

**Audience:**

Yes

**Broader Impact Concerns:**

Broader impact concerns are not identified from this paper.

**Claims And Evidence:**

Yes

**Requested Changes:**

•	Discuss the advantages of using PerSEval over other metrics in various contexts and use cases, and the impact of this new metric in this field. This will help readers understand the significance of this paper.

**Strengths And Weaknesses:**

Strengths

•	The necessary explanations provided as to why  EGISES and P-Acc fail to provide a reliable measure of personalization, justifying the necessity and motivation of this paper.

•	The thoroughness of including Accuracy-inconsistency Penalty (ACP) to account for corner cases where the best performance can be an outlier. Well done.

•	The evaluation framework that comprehensively assesses the reliability, stability, etc of  PerSEval, also involving human review is commendable.

•	The use of surrogate personalized summaries across multiple datasets to compensate for the limitations of dataset similar to PENS.

Weaknesses

•	Overall the paper is well-written, logical, and clear. I also read authors' answers on reviews and don't see big concerns. One thing that authors may consider adding is the user guide for applying this metric in different contexts, i.e. would there be any differences in optimizing PerSEval during model trainings comparing to traditional metrics?

---

> ### Author Response · Authors · 2024-09-07
> **Response to suggestions made by Reviewer j1ks**
>
> First, we would like to thank the reviewer for giving us valuable feedback within a short period of time. We are very happy and encouraged to see that the methodological details and measure design nuances were critically observed and appreciated. Regarding the additional suggestion given by the reviewer, we have noted that and made certain changes in the revised draft. The suggestion was:
>
> $\textit{\textcolor{red}{Regarding addition of user guideline explaining the use-cases and impact of PerSEval}}$ -- Thank you for pointing this out. We also felt this is much needed to put the overall contribution in the right perspective. We added a whole paragraph in the Discussion and Future Work section (marked in red).
>
> We look forward to a quick judgment on the paper.
>
> Best,
> Authors

---

### Comment · Reviewer_hE9R · 2024-07-18
**It seems that there is no author response**

Deal all,

It seems to be in the recommendation stage. However, I could not find any author response. I am wondering if there is a technical issue, or if the authors simply did not provide a response, in case I missed anything.

Thanks,
Reviewer hE9R

---

> ### Author Response · Authors · 2024-07-18
>
> Dear Reviewer hE9R,
>
> Thank you for your message.
>
> We are supposed to submit the revised version of our paper by today. Meanwhile, one of our leading authors of the paper got seriously sick and was admitted to the hospital. Therefore, we could not finish all the experiments. Today, we requested the EIC and AE to give us an extension of 7 days. We are yet to receive a response. We are committed to submitting our responses as soon as possible.
>
> Thank you very much for your consideration.

---

### Decision · Action_Editor_zBaZ · 2024-09-21

**Recommendation:** Accept as is

**Comment:**

The paper aims to improve improve evaluation of summarization, especially personalization aspect. The authors introduce a new metric PerSEval which is compared on 10+ state-of-the-art models and show high human-judgment correlation and rank stability, suggesting that the proposed metric is a more comprehensive and reliable measure of personalized summarization. The reviewers were leaning positive but had some concerns about (1) limited comparison with P-Acc, (2) dataset diversity (3) lack of computational considerations. In the revised version post rebuttals, the authors responded to all major concerns raised by the reviewers. They added more datasets to address concerns about the generality of their results and provided additional discussions on the limitations of P-Acc, showing that it cannot fully solve the issues related to personalization. As most of these concerns were resolved, it makes the paper ready for publication.

**Audience:**

Yes, a good portion of NLP community (especially long generation and summarization related) can be interested in this evaluation framework.

**Claims And Evidence:**

Mainly an evaluation paper. The authors conduct extensive experiments, including testing multiple state-of-the-art models and correlating the results with human judgments. The authors promise to opensource the code.